# VLG-CBM: Training Concept Bottleneck Models with Vision-Language Guidance

**Divyansh Srivastava**,* **Ge Yan**\*, **Tsui-Wei Weng**
{ddivyansh, geyan, lweng}@ucsd.edu
UC San Diego

## Abstract

Concept Bottleneck Models (CBMs) provide interpretable prediction by introducing an intermediate Concept Bottleneck Layer (CBL), which encodes human-understandable concepts to explain models' decision. Recent works proposed to utilize Large Language Models and pre-trained Vision-Language Models to automate the training of CBMs, making it more scalable and automated. However, existing approaches still fall short in two aspects: First, the concepts predicted by CBL often mismatch the input image, raising doubts about the faithfulness of interpretation. Second, it has been shown that concept values encode unintended information: even a set of random concepts could achieve comparable test accuracy to state-of-the-art CBMs. To address these critical limitations, in this work, we propose a novel framework called Vision-Language-Guided Concept Bottleneck Model (VLG-CBM) to enable faithful interpretability with the benefits of boosted performance. Our method leverages off-the-shelf open-domain grounded object detectors to provide visually grounded concept annotation, which largely enhances the faithfulness of concept prediction while further improving the model performance. In addition, we propose a new metric called Number of Effective Concepts (NEC) to control the information leakage and provide better interpretability. Extensive evaluations across five standard benchmarks show that our method, VLG-CBM, outperforms existing methods by at least 4.27% and up to 51.09% on *Accuracy at NEC=5* (denoted as ANEC-5), and by at least 0.45% and up to 29.78% on *average accuracy* (denoted as ANEC-avg), while preserving both faithfulness and interpretability of the learned concepts as demonstrated in extensive experiments[2].

## 1 Introduction

As deep neural networks become popular in real-world applications, it is crucial to understand the decision of these black-box models. One approach to provide interpretable decisions is the Concept Bottleneck Model (CBM) [6], which introduced an intermediate concept layer to encode human-understandable concepts. The model makes final predictions based on these concepts. Unfortunately, one major limitation of this approach is that it requires concept annotations from human experts, making it expensive and less applicable in practice as concept labels may not always be available.

Recently, a line of works utilized the powerful Vision-Language Models (VLMs) to replace manual annotation [15, 27, 25]. They used Large Language Models (LLMs) to generate set of concepts, and then trained the models in a post-hoc manner under the guidance of VLMs or neuron-level interpretability tool [14]. By eliminating the expensive manual annotations, some of these CBMs [15] could be scaled to large datasets such as ImageNet [18]. However, these CBMs [15, 27, 25] still face two critical challenges:

---

*Equal contribution

[2]Our code is available at https://github.com/Trustworthy-ML-Lab/VLG-CBM

38th Conference on Neural Information Processing Systems (NeurIPS 2024).

1. **Challenge #1: Inaccurate concept prediction.** The concept predictions in these CBMs often contain factual errors i.e. the predicted concepts do not match the image. Moreover, as concepts are generated by LLMs, there are some non-visual concepts, for example "loud music" or "location" used in LF-CBM [15], which further hurt the faithfulness of concept prediction.

2. **Challenge #2: Information leakage.** Recently, [13, 12] observed the information leakage in CBMs through empirical experiments – they found that the concept prediction encodes unintended information for downstream tasks, even if the concepts are irrelevant to the task.

In this paper, we propose a new framework called **V**ision-**L**anguage-**G**uided **C**oncept **B**ottleneck **M**odel (VLG-CBM) to address these two major challenges. Our contributions are summarized below:

1. To address **Challenge #1**, we propose to use the open-domain grounded object detection model to generate localized, visually recognizable concept annotations in Section 3. This approach automatically filters the non-visual concepts. Furthermore, the location information is utilized to augment the data. As far as we know, our VLG-CBM is the first end-to-end pipeline to build CBM with vision guidance from open-vocabulary object detectors.

2. To address **Challenge #2**, we provide the first rigorous theoretical analysis which proves that CBMs have serious issues on information leakage in Section 4.1, whereas previous study on information leakage [12, 25] only provides empirical explanations. Building on our theory, we further propose a new metric called the Number of Effective Concepts (NEC) in Section 4.2, which facilitates fair comparison between different CBMs. We also show that using NEC can help to effectively control information leakage and enhance interpretability in our VLG-CBM.

3. We conduct a series of experiments in Section 5 and demonstrate that our VLG-CBM outperforms existing methods across 5 standard benchmarks by at least 4.27% and up to 51.09% on *Accuracy at NEC=5* (denoted as ANEC-5), and by at least 0.45% and up to 29.78% on *average accuracy across different NECs* (denoted as ANEC-avg). Our learned CBM achieves a high sparsity of 0.2% in the final layer even on large datasets including Places365, preserving interpretability even with a large number of concepts. Additionally, we qualitatively demonstrate that our method provides more accurate concept attributions compared to existing methods.

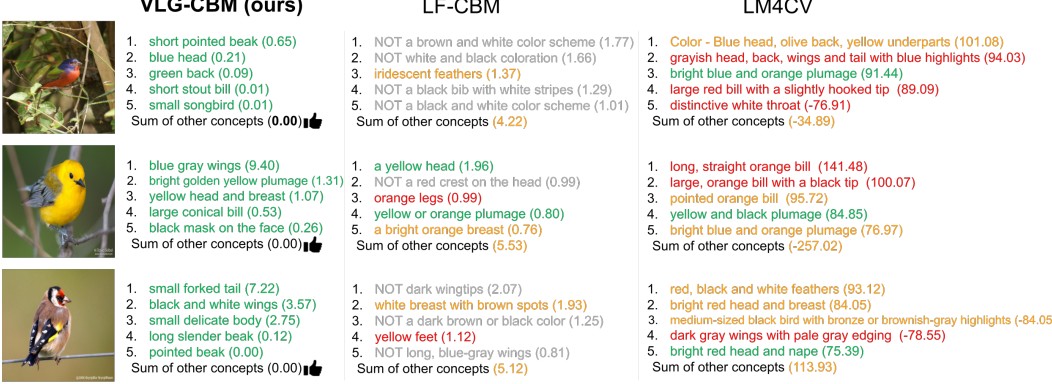

Figure 1: We compare the decision explanation of VLG-CBM with existing methods by listing top-5 contributions for their decisions. Our observations include: (1) VLG-CBM provides *concise* and *accurate* concept attribution for the decision; (2) LF-CBM [15] frequently uses negative concepts for explanation, which is less informative; (3) LM4CV[25] attributes the decision to concepts that do not match the images, a reason for this is that LM4CV uses a limited number of concepts, which hurts CBM's ability to explain diverse images; (4) Both LF-CBM and LM4CV have a significant portion of contribution from non-top concepts, making decisions less transparent. Full figure is in Appendix Fig. D.1.

| Method | Evaluation | | Flexibility | | Interpretability | |
|---|---|---|---|---|---|---|
| | Control on information leakage | Unlimited concept numbers | Flexible backbone | Accurate concept prediction | Vision-guided concept filtering | Interpretable decision |
| **Baselines:** | | | | | | |
| LF-CBM[15] | △ | ✓ | ✓ | △ | × | △ |
| LaBo[27] | × | ✓ | × | △ | × | △ |
| LM4CV[25] | ✓ | × | × | △ | △ | △ |
| **This work:** | | | | | | |
| VLG-CBM | ✓ | ✓ | ✓ | ✓ | ✓ | ✓ |

Table 1: Comparative analysis of methods based on evaluation, flexibility, and interpretability. Here, ✓ denotes the method satisfies the requirement, △ denotes the method partially satisfies the requirement, and × denotes the method does not satisfy the requirement. We compare with SOTA methods including LF-CBM [15], Labo [27] and LM4CV [25].

## 2    Related work

**Concept Bottleneck Model (CBM).** The seminal paper [1] first proposed self-explaining models by leveraging the idea of autoencoder to learn interpretable basis concepts in an unsupervised manner without pre-specified concepts. Later, [6] proposed to learn interpretable concepts with human specifications (labels) in the concept bottleneck layer (CBL), and coin the term Concept Bottleneck Models (CBM). CBL is followed by a linear prediction layer, which maps concepts to classes, enabling interpretable final decisions. Formally, let feature representation generated by a frozen backbone represented by $z = \phi(x)$, CBL concept prediction as $g(z) = W_c z$, and the final prediction layer as $h(\cdot) = W_F g(z) + b_F$. The final class prediction of the CBM is given by $\hat{y} = h(g(z)) = h \circ g \circ \phi(x)$.

Under this setting, the key in training a CBM is obtaining an annotated {(image, concept)} paired dataset for training concept bottleneck layer $g$. In [6], the authors used human-specified labels to train the CBL in a supervised way. However, obtaining labels with human annotators could be very tedious and costly. Recently, [15], [25], and [27] proposed to utilize Large Language Models (LLM) to generate a set of concepts $S$, then train CBL by aligning image and concepts with the guidance of vision language models (e.g. CLIP). For example, Oikarinen et al. [15] proposed LF-CBM to train CBM by directly learning a mapping from the embedding space of backbone to concept values in the CLIP space using cosine cubed loss function with the neuron interpretability tool[14], and then mapping concepts to classes using sparse linear layer. [25] proposed LM4CV, a task-guided concept searching method that learns text embeddings in the CLIP space, and then maps the learned embeddings to concepts obtained from LLM using nearest neighbor. Yang et al. [27] proposed LaBo, using submodular optimization to reduce the concept set, followed by using CLIP backbone for obtaining concept values. However, as we show in Sections 5.1 and 5.3 , these methods suffer from multiple issues: (i) The concept prediction is often incorrect and does not capture the visual attributes required for downstream class prediction (e.g. see Fig. 1 b )(ii) VLMs like CLIP suffer from modality gap between image and text embeddings [9] resulting in encoding unintended information, and even random concepts can achieve high accuracy [25]. To address these issues, we explicitly ground the concepts on the training dataset using an open-domain object detection model and then using the obtained concepts for learning CBL – this can ensure a more faithful representation of fine-grained concepts and avoids the modality gap issues introduced by VLMs. Table 1 demonstrates the superiority of VLG-CBM over existing methods [15, 25, 27] on properties including controlling information leakage, flexibility to use any backbone, and accurate concept prediction.

There are some recent works aim at addressing the challenges of CBMs. Similar to us, Pham et al. [16] uses an open-vocabulary object detection model to provide an explainable decision. However, their model is directly adapted from an OWL-ViT model, while our VLG-CBM uses an open-vocabulary object detection model to train a CBL over any base model, providing more flexibility. Additionally, their model requires pretraining to get best performance, while our VLG-CBM could be applied post-hoc to any pretrained model. Kim et al. [4] proposed to filter non-visual concepts by adding a vision activation term to the concept selection step, whereas VLG-CBM uses an open-vocabulary object detectors in multiple stage of CBM pipeline: for filtering non-visual concepts and the guiding training of concept bottleneck layer. Sun et al. [20] aims at eliminating the information leakage, and the authors evaluate the information leakage by measuring the performance drop speed after

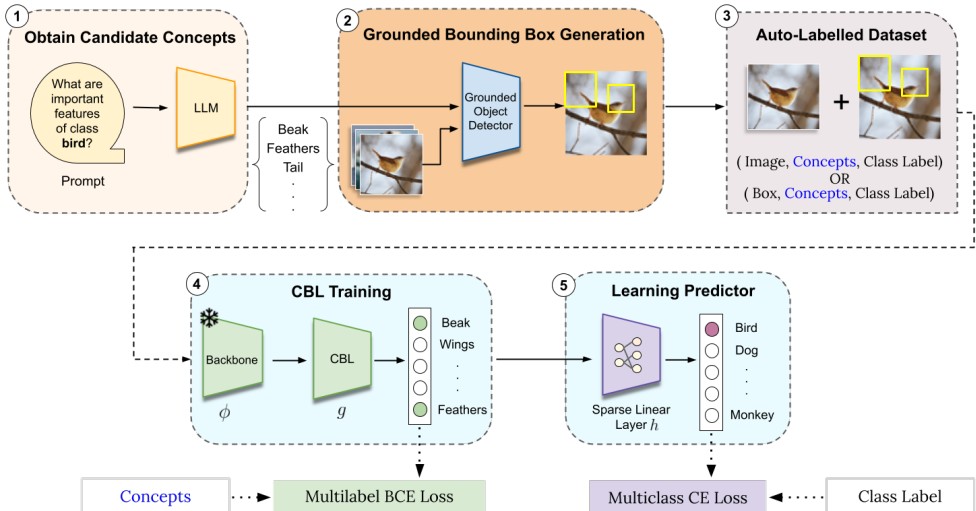

Figure 2: VLG-CBM pipeline: We design automated Vision+Language Guided approach to train Concept Bottleneck Models.

removing top-contributing concepts. This metric can be controlled by our proposed NEC metric, because the performance reach minimum after removing all contributing concepts. Roth et al. [17] demonstrate that random words and characters achieve comparable CLIP zero-shot performance on visual classification tasks. However, their work does not address information leakage problem and is a very different setting from our work. To date, most of the CBMs focused on vision domains, including this work. There are some recent work applying CBM approach to different domains and different tasks, e.g. interpretable language models for text classifications [21, 22, 11] and for continual learning [26]. We refer the interested readers to their papers for more details.

**Open Domain Language Grounded Object Detection.** Recent works, including GLIP [8], GLIPv2 [29], and GroundingDINO [10] detect objects in images in an open-vocabulary manner conditioned on natural language queries. In this work we propose to utilize open-vocabulary object detectors for automatically generating grounded concept dataset for training CBMs. This removes the need for human labelers, which is costly, tedious, and does not scale to large datasets. Further, the detected objects provide necessary vision-guidance for CBMs training as demonstrated in our experiments.

## 3 Method

In this section, we describe our novel automated approach to train a CBM with both Vision and Language Guidance to ensure faithfulness, which is currently lacking in the field. Our approach, abbreviated as VLG-CBM in the paper, generates an auxiliary dataset grounded on fine-grained concepts present in images for training a sequential CBM. Section 3.1 describes our approach to generating an auxiliary dataset used in training CBM, Section 3.2 describes our approach to training concept bottleneck layer, and Section 3.3 describes the training of sparse layer to obtain class labels from concepts in an interpretability-preservable manner. The overall pipeline is shown in Fig. 2.

### 3.1 Automated generation of auxiliary dataset

Here we describe our novel automated approach for generating labeled datasets for training CBMs. Let $f : \mathcal{X} \to \mathcal{Y}$ be the neural network mapping images to corresponding class labels, where $\mathcal{X} = \mathbb{R}^{H \times W \times 3}$ denotes the input image space and $\mathcal{Y} = \{1, 2, \ldots, C\}$ denotes the label space, $C$ is the number of classes. Denote $D = \{(x_i, y_i)\}, x_i \in \mathcal{X}, y_i \in \mathcal{Y}$ the dataset used for training $f$, where $x_i$ is the $i$-th image and $y_i$ is the corresponding label. Let $S$ be a set of natural-language concepts describing the fine-level visual details from which classes are composed. We propose to generate a modified and auxiliary dataset $D'$ from $D$ such that each image contains finer-grained concepts that

are useful in predicting the classes, along with the target class. The overall process of obtaining the modified dataset $D'$ can be divided into two steps:

- **Language supervision from LLMs to generate a set of candidate concepts**: We follow the steps proposed in LF-CBM [15] for generating candidate concepts $S_c$ for each class $c$ by prompting LLM to obtain visual features describing the class.

- **Vision supervision from Open-domain Object Detectors to ground candidate concepts to spatial information**: We propose using Grounding-DINO[10] Swin-B, current state-of-the-art grounded object detector, for obtaining bounding boxes of candidate concepts in the dataset. For each image $x_i$ with class label $c$ and candidate concepts $S_c$, we prompt Grounding DINO model with $S_c$ and obtain $K_i$ bounding boxes:

$$B_i = \{(b_j, t_j, s_j)\}_{j=1}^{K_i}, \tag{1}$$

where $b_j \in \mathbb{R}^{4 \times 2}$ is the $j$-th bounding box coordinates, $t_j \in \mathbb{R}$ is the corresponding confidence given by the model and $s_j \in S_c$ is the concept of this bounding box. We define a confidence threshold $T$ and remove bounding boxes with confidence less than $T$ to get filtered bounding boxes for each image:

$$\tilde{B}_i = \{(b, t, s) \in B_i \mid t > T\}. \tag{2}$$

After collecting bounding boxes for every image, we filter out the concepts that do not appear in any bounding box, and get our final concept set $\tilde{S}$:

$$\tilde{S} = \{s \in S \mid \exists(\cdot, \cdot, s) \in \cup_{i=1}^{|D|} \tilde{B}_i\}. \tag{3}$$

The one-hot encoded concept label vector $o_i \in \{0, 1\}^{|\tilde{S}|}$ for image $x_i$ is thus defined as:

$$(o_i)_j = \begin{cases} 1, & \text{if } s_j \text{ appears in } \tilde{B}_i, \\ 0, & \text{otherwise.} \end{cases} \tag{4}$$

Our final concept-labeled dataset $D'$ for training CBM can be written as:

$$D' = \{(x_i, o_i, y_i)\}_{i=1}^{|D|} \tag{5}$$

### 3.2 Training Concept Bottleneck Layer

After constructing the concept-labeled dataset $D'$, we now define our approach to train the concept bottleneck layer for predicting the fine-grained concepts in the input image in a multi-label classification setting. Let $\phi : \mathcal{X} \to \mathbb{R}^d$ be a backbone that generates $d$-dimensional embeddings $z = \phi(x)$ for input image $x$. Note that $\phi(x)$ can be a pre-trained backbone or trained from scratch. Define $g$ to be the Concept Bottleneck Layer (CBL) which maps embeddings to concept logits. We train a sequential CBM [6, 13] $g(\phi(x))$ to predict concepts in an image using Binary Cross Entropy (BCE) loss for multi-label prediction. Additionally, to improve the diversity of the concept-labeled dataset $D'$, we augment the training dataset by cropping images to a randomly selected bounding box and modifying the target one-hot vector to predict the concept corresponding to the bounding box. Our optimization objective in terms of BCE loss can be written as:

$$\min_g \mathcal{L}_{CBL}, \ \mathcal{L}_{CBL} = \frac{1}{|D'|} \sum_{i=1}^{|D'|} BCE[g \circ \phi(x_i), o_i] \tag{6}$$

### 3.3 Mapping Concept to Classes

In this section, we define our approach to training a sparse linear layer to obtain class labels from concepts in an interpretability-preservable manner. Let $h : \mathbb{R}^d \to \mathbb{R}^C$ be the sparse linear layer with weight matrix $W_F$ and bias $b_F$, which maps concept logits to class logits. We train the sparse layer using the original dataset $D$ by first obtaining concept logits from the trained CBL(frozen), normalizing the concept logits with the mean and variance on training set, and then using them to predict class logits. Our optimization objective in terms of Cross Entropy (CE) loss can be written as:

$$\min_h \mathcal{L}_{SL}, \ \mathcal{L}_{SL} = \frac{1}{|D|} \sum_{(x,y) \in D} CE[h \circ g \circ \phi(x), y] + \lambda R_\alpha, \tag{7}$$

where $R_\alpha = (1-\alpha)\frac{1}{2}\|W_F\|_2^2 + \alpha\|W_F\|_1$ is the elastic-net regularization [31] on weight matrix $W_F$, $\lambda$ is a hyperparameter controlling regularization strength. We use GLM-SAGA[24] solver to solve this optimization problem.

# 4 Unifying CBM evaluation with Number of Effective Concepts (NEC)

Besides training, another important challenge for CBM is: *how to evaluate the semantic information learned in the CBL?* Conventionally, the classification accuracy for final class labels is an important metric for evaluating CBMs, with the intuition that a good classification accuracy indicates that useful semantic information is learned in the CBL. However, purely using accuracy as the evaluation metric could be problematic, as it has been shown that information leakage exists in jointly or sequentially trained CBM [13, 12]. That is to say, the CBL could contain *unintended information* that could be used for downstream classification hence achieving high classification accuracy, even if the concept is irrelevant to the task. In fact, recently [25] showed that, when increasing the number of concepts, a randomly selected concept set could even approach the accuracy of the concept set chosen with sophistication, supporting the existence of information leakage.

To better understand this phenomenon, in section 4.1, we conduct a first theoretical analysis to investigate random CBL and its capability. To the best of our knowledge, this is the first formal analysis of random CBL. Next, inspired by our theoretical result, we propose a new evaluation metric for CBM, named NEC in section 4.2. NEC provides a way to control information leakage and enhance the interpretability of model decisions.

## 4.1 Theoretical analysis of the Random CBL

We start by defining the notations. Denote $k$ the number of concepts in CBL. We assume that the CBL $g$ consists of a single linear layer: $g(z) = W_c z$, where $W_c \in \mathbb{R}^{k \times d}$ and $z \in \mathbb{R}^d$, and the final layer $h$ is also linear: $h \circ g(z) = W_F g(z) + b_F$, where $W_F \in \mathbb{R}^{C \times k}$, $b_F \in \mathbb{R}^C$. This is the common setting for CBMs. The following theorem suggests a surprising conclusion: *a linear classifier upon random (i.e. untrained) CBL could accurately approximate any linear classifier trained directly on the representation, as the number of concepts in the CBL goes up.*

**Theorem 4.1.** *Suppose $\Sigma \in \mathbb{R}^{d \times d}$ is the variance matrix of the representation $z$ which is positive definite, $\lambda_{max}$ is the largest eigenvalue of $\Sigma$, and the weight matrix $W_c \in \mathbb{R}^{k \times d}$ is sampled i.i.d from a standard Gaussian distribution. For any linear classifier $f$ which is built directly on the representation $z$, i.e. $f : \mathbb{R}^d \to \mathbb{R}, f(z) = w^\top z + b$, it could be approximated by another linear classifier $\tilde{f}$ on concept logits $g(z) = W_c z$, i.e. $f(z) \approx \tilde{f}(z) = \tilde{w}^\top g(z) + \tilde{b}$, with the expected square error $E(k)$ upper-bounded by*

$$E(k) \le \begin{cases} \lambda_{max}(1 - \frac{k}{d})\|w\|_2^2, & k < d; \\ 0, & k \ge d. \end{cases} \tag{8}$$

*Here $E(k) = \mathbb{E}_{W_c}\left[\min_{(\tilde{w},\tilde{b})} \mathbb{E}_z \left[|f(z) - \tilde{f}(z)|^2\right]\right]$ denotes the average square error, $w \in \mathbb{R}^d$, $\tilde{w} \in \mathbb{R}^k$, $k$ is the number of concepts in CBL.*

*Remark* 4.1. In Theorem 4.1, we consider a 1-D regression problem where we use a linear combination of concept bottleneck neurons to approximate any linear function. The multi-class classification result could be derived by applying Theorem 4.1 to each class logit (see Corollary A.1). From Eq. (8), we could see that the expected error goes down linearly when concept number $k$ increases, and achieves 0 when $k \ge d$, where $d$ is the dimension of backbone representation $z$. This suggests that, even with a random CBL (i.e. $W_c$ is simply drawn from a standard Gaussian distribution without any training), the classifier could still approximate the original classifier well and achieve good accuracy, when concept number $k$ is large enough. We defer the formal proof of Theorem 4.1 to Appendix A.

## 4.2 A New Evaluation Metric for CBM: Number of Effective Concepts (NEC)

Theorem 4.1 provides a formal theoretical explanation on [25]'s observation. Moreover, it raises a concern on the evaluation of CBMs: *model classification accuracy may not be a good metric for evaluating the semantical information learned in CBL, because a random CBL could also achieve high accuracy.* To address this concern, we need to control the concept number $k$ so that the

semantically meaningful CBLs can be distinguished with random CBLs w.r.t. the final classification performance.

We notice that previous works mainly use two approaches to control $k$:

1. Control the total number of concepts: [25] used a more concise concept layer, i.e. reduce the total number of concepts. However, this approach may miss some important concepts due to limitations in total concept numbers. Additionally, they used a dense final layer which is less interpretable for humans, as each decision is related to the whole concept set.

2. [15, 28] suggested using a sparse linear layer for final prediction to enhance interpretability. Though sparsity is initially introduced to enhance interpretability, we note that this also reduces the number of concepts used in the decision, thus controlling the information leakage. However, the problem is, these works lack the quantification for sparsity, which is necessary for fair comparison between methods.

To provide a unified metric for both approaches, we propose to measure the Number of Effective Concepts (NEC) for final prediction as a sparsity metric. It is defined as

$$NEC(W_F) = \frac{1}{C} \sum_{i=1}^{C} \sum_{j=1}^{k} \mathbf{1}\{(W_F)_{ij} \neq 0\} \tag{9}$$

Intuitively, NEC measures the average number of concepts the model uses to predict a class. Using NEC to evaluate CBM provides the following benefits:

1. A smaller NEC reduces the information leakage. As shown in Fig. 3, with large NEC, even random CBL could achieve near-optimal accuracy, suggesting potential leakage in information. However, by reducing NEC, the accuracy of random concepts drops quickly. This implies enforcing a small NEC could help to control information leakage.

2. A model with a smaller NEC provides more interpretable decision explanations. Humans can recognize an object with several important visual features. However, models can utilize tens or hundreds of concepts for the final prediction. By using a smaller NEC, the model's decision could be attributed mainly to several concepts, making it more interpretable to human users.

3. NEC enables fair comparison between CBMs. Comparing the performance of CBMs has long been a challenging problem, as different models use different numbers of concepts and different styles of final layers (sparse/dense). NEC considers both, thus providing a fair metric to compare different models.

Given these benefits, we suggest to control the NEC when comparing the performance of CBMs. In Section 5, we provide experiments with controlled NEC, where we observed our VLG-CBM outperforms other baselines.

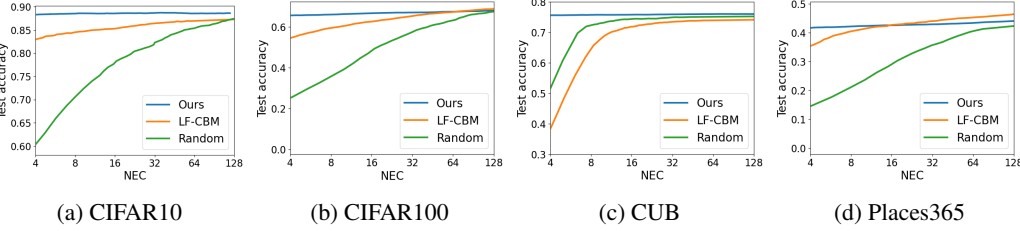

| (a) CIFAR10 | (b) CIFAR100 | (c) CUB | (d) Places365 |

Figure 3: Accuracy comparison between our VLG-CBM, LF-CBM[15] and randomly initialized concept bottleneck layer under different NEC. The experiment is conducted on the CIFAR10 dataset. From the results, we could see that (1) for NEC large enough, even a random CBL could achieve near-optimal accuracy, supporting the existence of information leakage; (2) when NEC decreases, the accuracy of LF-CBM and random weights begin to drop, while our VLG-CBM does not have significant decrease.

# 5 Experiments

In this section, we conduct a series of experiments to evaluate our method, including illustrating the faithfulness of concept prediction, interpretability of model decisions, and performance with controlled NEC.

## 5.1 Performance comparison

**Setup.** Following prior work [15], we conduct experiments on five image recognition datasets: CIFAR10, CIFAR100[7], CUB[23], Places365[30] and ImageNet[18]. For the choice of backbone, we categorize the experiments into two categories:

1. CLIP backbone: For CLIP backbone, we choose CLIP-RN50 for all datasets.
2. non-CLIP backbobe: We follow the choice of [15] to use ResNet-18[3] for CUB and ResNet-50 (trained on ImageNet) for Places365 and ImageNet as the backbone.

The reason of this categorization is some previous works (e.g. LaBo[27] only supports CLIP backbone.

**Baselines.** We compare our method with three major baselines when applicable: LF-CBM[15], LaBo[27], and LM4CV[25]. These are SOTA methods for constructing scalable CBMs, with [15] most flexible and [27, 25] limited by specific architecture and not available for certain dataset. Additionally, we present the results from a randomly initialized CBL for comparison.

**Metrics.** As discussed in Section 4, in order to evaluate the final classification power of CBM, we should acculate the Accuracy under specified NEC(ANEC). Therefore, we measure the following two metrics:

1. **Accuracy at NEC=5(ANEC-5):** This metric is designed to show the performance of CBM which could provide an interpretable prediction. We choose the number 5 so that human users could easily inspect all concepts related to the decision without much effort.
2. **Average accuracy(ANEC-avg):** To evaluate the trade-off between interpretability and performance, we also calculate the average accuracy under different NECs. In general, higher NEC indicates a more complex model, which may achieve better performance but also hurt interpretability. We choose six different levels: $5, 10, 15, 20, 25, 30$ and measure the average accuracy.

**Controlling NEC.** As we discussed in Section 4, there are two approaches to control NEC: (1) using a dense final layer and directly controlling the number of concepts and (2) training a sparse final layer with appropriate sparsity. LM4CV[25] used the first approach, where the number of concepts could be directly set as target NEC. For LF-CBM[15] and our VLG-CBM, the second approach is utilized: To achieve target sparsity, we control the regularization strength $\lambda$ in GLM-SAGA. GLM-SAGA provides a regularization path, which allows us to gradually reduce regularization strength and get a series of weight matrices with different sparsity. Specifically, we start with $\lambda_0 = \lambda_{max}$ which gives the sparsest weight. Then, we gradually reduce the $\lambda$ by $\lambda_{t+1} = \alpha\lambda_t$ until we achieve the desired NEC. We choose the weight matrix with the closest NEC to our target and prune the weights from smallest magnitude to largest to enforce accurate NEC. LaBo [27] did not provide a NEC control method. Hence, we apply sparse final layer training of LaBo's concept prediction to control NEC.

**Results.** We summarize the test results for with backbone in Table 2 and results with non-CLIP backbones in Table 3. From the results, we could see that:

1. The accuracy at NEC $= 5$ provides a good metric for evaluating the semantic information in CBL: As shown in the table, the accuracy of random CBL is much lower with NEC $= 5$, which implies the information leakage is controlled and the accuracy could better reflect the useful semantic information learned in the CBL.
2. The performance of LM4CV is even worse than random CBL. An explanation to this is LM4CV utilizes a dense final layer, which is intrinsically inefficient to interpret as each class is connected to all the concepts, including the irrelevant ones. When limiting the NEC to a small value to provide a concise explanation, the model has to largely reduce the concept number which sacrifices the prediction power.

3. Our method significantly outperforms all the baselines at least 4.27% and up to 51.09% on accuracy at NEC=5, and by at least 0.45% and up to 29.78% on average accuracy across different NECs, illustrating both high performance and good interpretability.

| Dataset | CIFAR10 | | CIFAR100 | | ImageNet | | CUB | |
|---|---|---|---|---|---|---|---|---|
| Metrics | ANEC-5 | ANEC-avg | ANEC-5 | ANEC-avg | ANEC-5 | ANEC-avg | ANEC-5 | ANEC-avg |
| Random | 67.55% | 77.45% | 29.52% | 47.21% | 18.04% | 39.63% | 25.37% | 40.13% |
| LF-CBM | 84.05% | 85.43% | 56.52% | 62.24% | 52.88% | 62.24% | 31.35% | 52.70% |
| LM4CV | 53.72% | 69.02% | 14.64% | 36.70% | 3.77% | 26.65% | 3.63% | 15.25% |
| LaBo | 78.69% | 82.05% | 44.82% | 55.18% | 24.27% | 45.53% | 41.97% | 59.27% |
| **VLG-CBM(Ours)** | **88.55%** | **88.63%** | **65.73%** | **66.48%** | **59.74%** | **62.70%** | **60.38%** | **66.03%** |

Table 2: Performance comparison with CLIP-RN50 backbone. We compare our method with a random baseline, LF-CBM[15], LM4CV[25] and LaBo[27]. The random baseline has 1024 neurons for CIFAR10 and CIFAR100, 512 for CUB and 4096 for ImageNet.

| Dataset | CUB | | Places365 | | ImageNet | |
|---|---|---|---|---|---|---|
| Metrics | ANEC-5 | ANEC-avg | ANEC-5 | ANEC-avg | ANEC-5 | ANEC-avg |
| Random | 68.91% | 73.44% | 17.57% | 28.62% | 41.49% | 61.97% |
| LF-CBM | 53.51% | 69.11% | 37.65% | 42.10% | 60.30% | 67.92% |
| **VLG-CBM (Ours)** | **75.79%** | **75.82%** | **41.92%** | **42.55%** | **73.15%** | **73.98%** |

Table 3: Performance comparison with non-CLIP backbone. We compare against LF-CBM[15] and a random baseline, as LM4CV[25], LaBo[27] does not support non-CLIP backbone. The random baseline has 512 neurons for CUB, 2048 for Places365, and 4096 for ImageNet.

## 5.2 Visualization of CBL neurons

In order to examine whether our CBL learns concepts aligned with human perception, we list the top-5 activated images for example concept neurons on the model trained on the CUB dataset in Fig. 4. As shown in the figure, our CBL faithfully captures the corresponding concept. We provide more visualization results in Appendix K.

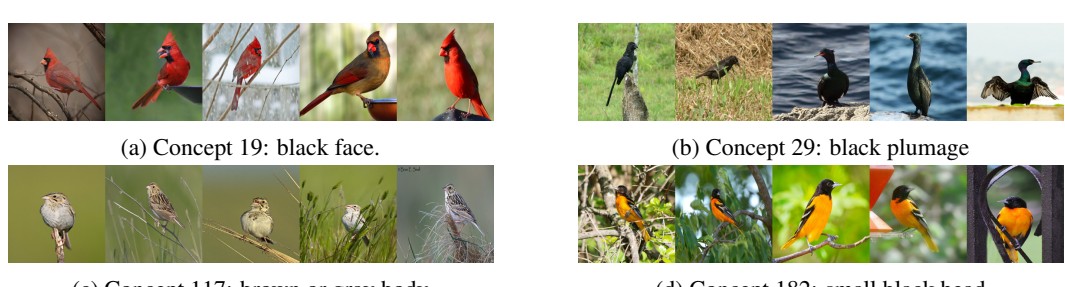

(a) Concept 19: black face.

(b) Concept 29: black plumage

(c) Concept 117: brown or gray body

(d) Concept 182: small black head

Figure 4: Top-5 activated images of example concepts neurons in VLG-CBM on CUB dataset.

## 5.3 Case study

In this section, we conduct a case study to compare the concept prediction between our VLG-CBM , LF-CBM [15] and LM4CV[25] as shown in Fig 1. We provide extended results and comparison with LaBo [27] in Appendix G.2. For our method, we use the final layer with NEC = 5. We show that our method provides more accurate concept prediction and more interpretable decisions for users.

**Decision interpretability.** We examine the explanation of each CBM model on example images by showing the top-5 concept contributions. The contribution of each concept is calculated as the product of the concept prediction value and corresponding weight. Formally, the contribution of

$i$-th concept to $j$-th class is defined as: $\text{Contribution}(i, j) = g_i(z) \cdot (W_F)_{ji}$. We pick the top-5 contributing concepts for the final predicted class and visualize it in Fig. 1. We could see that:

1. For other CBMs, a large portion of the final decision is attributed to the "Sum of other features". This part hurts the interpretability of CBM, as it's difficult for users to manually inspect all these concepts in practice. We conduct further study on this in Section 5.4. Our model, however, provides a concise explanation from a few concepts because we apply the constraint NEC=5. This ensures users can understand model decisions without difficulties.

2. Our VLG-CBM provides explanation more aligned with human perception. From the example, we can also see that our model explains the decision with clear visual concepts. Other CBMs attribute the decision to non-visual concepts (e.g. LaBo), concepts that do not match the image (e.g. LM4CV), or negative concepts (LF-CBM).

### 5.4  Do Top-5 concepts fully explain the decision?

Besides training a final layer with a small NEC, another common approach to provide a concise explanation is showing only the top contribution concepts. However, we argue that this approach may not faithfully explain the model's behavior, as the non-top concepts also make a significant contribution to the decision. To verify this, we conduct the following experiment: On the CUB dataset, we prune the final weight matrix $W_F$ to leave only the top-5 concepts for each class, whose weight has the largest magnitude. Then, we use the pruned model to make predictions and compare them with the prediction results from the original model. Table 4 shows results for our VLG-CBM which uses NEC$= 5$ to control sparsity, and other three baselines, LaBo[27], LF-CBM[15] and LM4CV[25], without any constraint on NEC. As shown in the table, for all three baselines, a large portion of predictions changes after pruning. This suggests that without explicitly controlling NEC, only showing top-5 contributing concepts does not faithfully explain all of the model decisions. Hence, we recommend training the final layer with NEC controlled to obtain a concise and faithful explanation as we proposed in Section 4.

| Method | VLG-CBM (Ours) | LF-CBM | LM4CV | LaBo |
|---|---|---|---|---|
| % changed decisions | **0.12%** | 49.21% | 98.34% | 81.40% |

Table 4: Portion of model predictions that changes after pruning. The results suggest that for existing methods (LF-CBM, LM4CV, LaBo) without NEC control, a large portion of predictions changes with top-5 concepts, implying potential risk when using top-5 contributions to explain model decisions.

## 6  Conclusion, Potential Limitations and Future work

In this work, we study how to improve the interpretability and performance of concept bottleneck models. We introduce a novel approach VLG-CBM based on both vision and language guidance, which successfully improves both the interpretability and utility of existing CBMs in prior work. Additionally, our theoretical analysis show that information leakage may exist on even in the untrained CBLs, serving the foundations for our proposed new NEC-controlled metrics (ANEC-5 and ANEC-avg). We show that NEC not only allow fair evaluation of CBMs but also can be used to effectively control information leakage of CBM and ensure interpretability. Extensive experiments on image classification benchmarks demonstrated our VLG-CBM largely outperform previous baselines especially for small NEC, providing more interpretable decisions for users.

Despite the superior performance of VLG-CBM over prior work as demonstrated in extensive experiments, one potential limitation is the dependence on large pretrained models (e.g. the success of open-domain grounded object detection model that we use to enforce vision guidance). However, prior work (e.g. LaBo, LM4CV, LF-CBM) also shared similar limitation on the reliance of large pre-trained models (e.g. CLIP). Nevertheless, it also means that our techniques have the potential to be further improved with the advancement of large pre-trained models. In the future, we plan to explore training CBL with even more vision guidance, such as using segmentation maps of concepts.

# Acknowledgement

The authors thank the anonymous reviewers for valuable feedback on the manuscript. The authors are partially supported by National Science Foundation awards CCF-2107189, IIS-2313105, IIS-2430539, Hellman Fellowship, and Intel Rising Star Faculty Award. The authors also thank ACCESS for support in this work.

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

# Appendix

## Table of Contents

# A  Proof of Theorem 4.1

In this section, we present a formal definition of the expected square error in Theorem 4.1 and show the proof. First, we define the square approximation error as

$$\mathbb{E}_z \left[ |f(z) - \tilde{f}(z)|^2 \right], \tag{A.1}$$

which is the average square distance between $f(z)$ and $\tilde{f}(z)$. Given a specific CBL $W_c$, we seek a final layer $\tilde{w}$ to minimize the square error:

$$\min_{(\tilde{w}, \tilde{b})} \mathbb{E}_z \left[ |f(z) - \tilde{f}(z)|^2 \right]. \tag{A.2}$$

For randomly Gaussian initialized $W_c$, we care about the minimal error we could achieve on average. Thus, for each $W_c$, we choose $\tilde{w}$ and $\tilde{b}$ to achieve minimum approximation error, then take the expectation over $W_c$ to define the expected square error as

$$E(k) = \mathbb{E}_{W_c} \left[ \min_{(\tilde{w}, \tilde{b})} \mathbb{E}_z \left[ |f(z) - \tilde{f}(z)|^2 \right] \right]. \tag{A.3}$$

**Setting**  Suppose the representation $z$ has variance $\Sigma \in \mathbb{R}^{d \times d}$ which is positive definite. The weight matrix $W_c \in \mathbb{R}^{k \times d}$ is sampled i.i.d from a standard Gaussian distribution. Here, we show that any linear classifier which is built directly on representation $z$, i.e. $f(z) = w^\top z + b$, could be approximated by a linear classifier on concept logits $g(z) = W_c z$, i.e. $f(z) \approx \tilde{f}(z) = \tilde{w}^\top g(z) + \tilde{b}$.

**Theorem 4.1.** *Suppose $\Sigma \in \mathbb{R}^{d \times d}$ is the variance matrix of the representation $z$ which is positive definite, $\lambda_{max}$ is the largest eigenvalue of $\Sigma$, and the weight matrix $W_c \in \mathbb{R}^{k \times d}$ is sampled i.i.d from a standard Gaussian distribution. For any linear classifier $f$ which is built directly on the representation $z$, i.e. $f : \mathbb{R}^d \to \mathbb{R}, f(z) = w^\top z + b$, it could be approximated by another linear classifier $\tilde{f}$ on concept logits $g(z) = W_c z$, i.e. $f(z) \approx \tilde{f}(z) = \tilde{w}^\top g(z) + \tilde{b}$, with the expected square error $E(k)$ upper-bounded by*

$$E(k) \le \begin{cases} \lambda_{max}(1 - \frac{k}{d})\|w\|_2^2, & k < d; \\ 0, & k \ge d. \end{cases} \tag{8}$$

*Here $E(k) = \mathbb{E}_{W_c} \left[ \min_{(\tilde{w}, \tilde{b})} \mathbb{E}_z \left[ |f(z) - \tilde{f}(z)|^2 \right] \right]$ denotes the average square error, $w \in \mathbb{R}^d$, $\tilde{w} \in \mathbb{R}^k$, $k$ is the number of concepts in CBL.*

*Proof.*  Based on the value of $k$, we can consider two cases: (I) $k < d$, and (II) $k \ge d$, and derive the $E(k)$ respectively.

**Case (I): $k < d$. First, we consider under a fixed $W_c$, what is the minimum error we could achieve.** The expected approximation error is:

$$\begin{aligned} \mathbb{E}_z \left[ |f(z) - \tilde{f}(z)|^2 \right] &= \mathbb{E}_z \left[ |w^\top z + b - (\tilde{w}^\top W_c z + \tilde{b})|^2 \right] \\ &= \mathbb{E}_z \left[ |(w^\top - \tilde{w}^\top W_c)z + b - \tilde{b}|^2 \right] \\ &= \underbrace{\mathbb{V}_z[(w - W_c^\top \tilde{w})^\top z]}_{(*)} + \underbrace{\left[ \mathbb{E}_z[(w^\top - \tilde{w}^\top W_c)z] + b - \tilde{b} \right]^2}_{(**)}. \end{aligned} \tag{A.4}$$

The last equality is from $\mathbb{E}(X^2) = \mathbb{V}X + (\mathbb{E}X)^2$. The second term $(**)$ takes minimum 0 when $\tilde{b} = \mathbb{E}_z[(w^\top - \tilde{w}^\top W_c)z] + b$. The remaining question is to choose a proper $\tilde{w}$ to minimize $(*)$. Notice that

$$\begin{aligned} \mathbb{V}_z \left[ (w - W_c^\top \tilde{w})^\top z \right] &= (w - W_c^\top \tilde{w})^\top \Sigma (w - W_c^\top \tilde{w}) \\ &= \|\Sigma^{\frac{1}{2}} (w - W_c^\top \tilde{w})\|_2^2 \\ &= \|\Sigma^{\frac{1}{2}} W_c^\top \tilde{w} - \Sigma^{\frac{1}{2}} w\|_2^2, \end{aligned} \tag{A.5}$$

where $\Sigma$ is the covariance matrix of $z$, $\Sigma^{\frac{1}{2}}$ is the principal square root of $\Sigma$, $\Sigma \in \mathbb{R}^{d \times d}$, $\Sigma^{\frac{1}{2}} \in \mathbb{R}^{d \times d}$. Now the problem in Eq. (A.2) can be reduced to a linear least square problem:

$$\min_{(\tilde{w},\tilde{b})} \mathbb{E}_z \left[ |f(z) - \tilde{f}(z)|^2 \right] = \min_{\tilde{w}} \|\Sigma^{\frac{1}{2}} W_c^\top \tilde{w} - \Sigma^{\frac{1}{2}} w\|_2^2 \tag{A.6}$$

Since $\Sigma$ is positive definite, so is $\Sigma^{\frac{1}{2}}$. Thus, the eigen decomposition of $\Sigma^{\frac{1}{2}}$ satisfies the following: $\Sigma^{\frac{1}{2}} = \tilde{Q}^\top \Lambda \tilde{Q}$, where $\tilde{Q} \in \mathbb{R}^{d \times d}$ is an orthogonal matrix and $\Lambda \in \mathbb{R}^{d \times d}$ is a diagonal matrix with positive entries. With Gram–Schmidt process, we could derive QR factorization of $W_c^\top$: $W_c^\top = QR$, where $Q \in \mathbb{R}^{d \times d}$ is orthogonal and $R \in \mathbb{R}^{d \times k}$ is upper triangular. Plugging above decomposition of $\Sigma^{\frac{1}{2}}$ and $W_c^\top$, now we have

$$
\begin{aligned}
\min_{\tilde{w}} \|\Sigma^{\frac{1}{2}} W_c^\top \tilde{w} - \Sigma^{\frac{1}{2}} w\|_2^2 &= \min_{\tilde{w}} \|\tilde{Q}^\top \Lambda \tilde{Q} Q R \tilde{w} - \tilde{Q}^\top \Lambda \tilde{Q} w\|_2^2 \\
&= \min_{\tilde{w}} \|\Lambda \tilde{Q} Q R \tilde{w} - \Lambda \tilde{Q} w\|_2^2 && (\tilde{Q} \text{ is orthogonal, thus preserves 2-norm}) \\
&= \min_{\tilde{w}} \|\Lambda (\tilde{Q} Q R \tilde{w} - \tilde{Q} w)\|_2^2 \\
&\leq \min_{\tilde{w}} \lambda_{max}^2 \|\tilde{Q} Q R \tilde{w} - \tilde{Q} w\|_2^2 && (\text{Since all entries of } \Lambda \text{ are positive.}) \\
&= \lambda_{max}^2 \min_{\tilde{w}} \|R \tilde{w} - Q^\top w\|_2^2 && (\text{Multiply by } Q^\top \tilde{Q}^\top \text{ preserves the norm})
\end{aligned}
$$
$$\tag{A.7}$$

where $\lambda_{max}$ is the largest eigenvalue of $\Sigma^{\frac{1}{2}}$. In short, we have derived the minimum square error for a given $W_c$, which is upper bounded by

$$\min_{(\tilde{w},\tilde{b})} \left[ \mathbb{E}_z \left[ |f(z) - \tilde{f}(z)|^2 \right] \right] \leq \lambda_{max}^2 \min_{\tilde{w}} \|R \tilde{w} - Q^\top w\|_2^2 \tag{A.8}$$

**Secondly, we consider when $W_c$ is sampled i.i.d. from standard normal distribution, and calculate the expected error.** From above derivation,

$$\mathbb{E}_{W_c} \left[ \min_{(\tilde{w},\tilde{b})} \mathbb{E}_z \left[ |f(z) - \tilde{f}(z)|^2 \right] \right] \leq \lambda_{max}^2 \mathbb{E}_{(R,Q)} \left[ \min_{\tilde{w}} \|R \tilde{w} - Q^\top w\|_2^2 \right] \tag{A.9}$$

Note that since $W_c^\top = QR$, the randomness in $W_c$ is reflected in $Q$ and $R$. The matrices $Q$ and $R$ satisfies the following properties:

1. $Q$ is a random rotation following uniform distribution. This is intuitive because standard Gaussian distribution is rotation-invariant. For a formal statement and proof, we refer to Proposition 7.2 of Eaton [2].

2. $range(R) = span(e_1, e_2, \cdots, e_k)$ with probability 1. Since $rank(R) = rank(W_c^\top)$ and $W_c^\top$ is full rank with probability 1, $rank(R) = \min(k, d) = k$ with probability 1. From upper-triangularity of $R$, we know that

$$range(R) \subseteq span(e_1, e_2, \cdots, e_k). \tag{A.10}$$

With probability 1, $rank(R) = k$, thus we conclude

$$range(R) = span(e_1, e_2, \cdots, e_k). \tag{A.11}$$

In the following derivation, since we only cares about the expectation, we omit "with probability 1" for brevity.

From the above properties of $Q$ and $R$, the expectation term in the RHS of Eq. (A.9) can be derived as:

$$\mathbb{E}_{(R,Q)} \left[ \min_{\tilde{w}} \|R \tilde{w} - Q^\top w\|_2^2 \right] = \mathbb{E}_Q \|(Q^\top w)_{k+1:d}\|_2^2. \tag{A.12}$$

This is because $range(R) = span(e_1, e_2, \cdots, e_k)$, and $k < d$. Thus, $\min_{\tilde{w}} \|R \tilde{w} - Q^\top w\|_2^2$ is the squared distance from $Q^\top w$ to subspace $span(e_1, e_2, \cdots, e_k)$, which equals to the squared sum of last $d - k$ coordinates of $Q^\top w$.

Because $Q$ is a random rotation, $Q^\top w$ is uniformly distributed on a sphere with radius $\|w\|_2$. Denote $v = Q^\top w$. From symmetricity, we have

$$\mathbb{E}\, v_1^2 = \mathbb{E}\, v_2^2 = \cdots = \mathbb{E}\, v_d^2. \tag{A.13}$$

Furthermore, $\|v\|_2^2 = \|w\|_2^2$ gives $\sum_{i=1}^d v_i^2 = \|w\|_2^2$. Take expectation of both sides gives $\sum_{i=1}^d \mathbb{E}_v\, v_i^2 = \|w\|_2^2$, thus $\mathbb{E}\, v_1^2 = \mathbb{E}\, v_2^2 = \cdots = \mathbb{E}\, v_d^2 = \|w\|_2^2/d$. The target quantity becomes

$$
\begin{aligned}
\mathbb{E}_Q \|(Q^\top w)_{k+1:d}\|_2^2 &= \mathbb{E}_v \left( \sum_{i=k+1}^d v_i^2 \right) \\
&= \sum_{i=k+1}^d \mathbb{E}_v\, v_i^2 \\
&= \frac{d-k}{d} \|w\|_2^2
\end{aligned}
\tag{A.14}
$$

In conclusion, we derive an upper bound of approximation error for any linear function $f$:

$$\mathbb{E}_{W_c} \left[ \min_{(\tilde{w}, \tilde{b})} \mathbb{E}_z \left[ |f(z) - \tilde{f}(z)|^2 \right] \right] \leq \lambda_{max}^2 (1 - \frac{k}{d}) \|w\|_2^2 \tag{A.15}$$

Look at the bound in Eq. (A.15): $\lambda_{max}^2$ is a constant regarding the scale of data; $\|w\|_2^2$ is a constant regarding the norm of weight vector we want to approximate; $(1 - \frac{k}{d})$ is a linear term shows that the expected square error goes down linearly when we increase the number of concepts $k$, and achieves zero when $k = d$.

**Case (II):** $k \geq d$. For the case that $k \geq d$, it could be derived from our main results that $E(k) = 0$. Additionally, with probability 1 we could find $\tilde{f}(x) = f(x)$ as will be derived below. As we discussed, with probability 1, $W_c$ has full rank. Given that, we have

$$W_c^+ W_c z = z,$$

where $W_c^+$ is the Moore-Penrose inverse of $W_c$. For any linear classifier $f(z) = w^\top z + b$. Let $\tilde{w} = (W_c^+)^\top w$, $\tilde{b} = b$, we have

$$\tilde{f}(z) = \tilde{w}^\top g(z) + \tilde{b} = w^\top W_c^+ W_c z + b = w^\top z + b = f(z)$$

and thus $E(k) = 0$. $\qquad\qquad\square$

**Corollary A.1.** *For $f$ and $\tilde{f}$ with $C$ output classes, i.e. $f : \mathbb{R}^d \to \mathbb{R}^C$, $\tilde{f} : \mathbb{R}^d \to \mathbb{R}^C$, $w \in \mathbb{R}^d$, $\tilde{w} \in \mathbb{R}^k$, the expected error upper-bound is*

$$E(k) \leq C\lambda_{max}(1 - \frac{k}{d})\|w\|_2^2. \tag{A.16}$$

*Here $E(k) = \mathbb{E}_{W_c} \left[ min_{(\tilde{w}, \tilde{b})} \mathbb{E}_z \|f(z) - \tilde{f}(z)\|^2 \right]$ denotes the average square error.*

*Remark* A.2. The statement could be verified by applying Theorem 4.1 to each $f_i$ and $\tilde{f}_i$ output, then summing up the error.

# B Implementation details

**Computational resources and codes.** Our experiments run on a server with 10 CPU cores, 64 GB RAM, and 1 Nvidia 2080Ti GPU. Our implementation builds on the open-source implementation of the LF-CBM [15] available: `https://github.com/Trustworthy-ML-Lab/Label-free-CBM`. For training the final predictive layer, we use publicly available code for GLM-SAGA [24].

**Hyperparameter tuning.** We tune the hyperparameters for our method using 10% of the training data as validation for the CIFAR10, CIFAR100, CUB and ImageNet datasets. For Places365, we use 5% of the training data as validation. We use CLIP(RN50) image encoder as the backbone for CIFAR10 and CIFAR100, Resnet-18[3] trained on CUB for CUB dataset, and Resnet-50 pretrained for Places365 following setup similar to LF-CBM. We tune the CBL with Adam[5] optimizer with learning rate $1\times10^{-4}$ and weight decay $1\times10^{-5}$. The concept dataset obtained from GroundingDINO is inherently unbalanced since there is a much lower proportion of positive datapoints for a concept. Consequently, we scale the CBL loss by multiplying it with a positive value to balance the tradeoff between precision and recall and improve the imbalance of positive data points. We set $T = 0.15$ in Eq. (2) in all our experiments. We seed the random number generator with a fixed seed to ensure the results can be reproduced.

# C Ablation Studies

| Confidence threshold | CUB200 | | Places365 | |
|:---:|:---:|:---:|:---:|:---:|
| Metrics | ANEC-5 | ANEC-avg | ANEC-5 | ANEC-avg |
| 0.10 | 75.75% | 75.75% | 41.84% | 42.50% |
| 0.15 | 75.75% | 75.73% | 41.84% | 42.51% |
| 0.20 | 75.73% | 75.73% | 41.25% | 42.15% |

Table C.1: ANEC-5 and ANEC-avg for different confidence threshold $T$.

## C.1 Ablation study for confidence threshold

Confidence threshold $T$ in Eq 2 filters concepts with bounding boxes' confidence less than $T$. In this experiment, we study the affect of T on the VLG-CBM's accuracy. The results are shown in Table C.1. We observe that ANEC-5 and ANEC-avg first increases (or stays constant) and then decreases. We attribute this effect to to the fact that as T increases, the number of false-positive decreases leading to better learning of concepts, however, as the number of annotations available for learning a concept decreases.

# D    Evaluating annotations from Grounding DINO

This section quantitatively evaluates concept annotations obtained from Grounding DINO. We use CUB dataset for comparison which contains ground-truth for fine-grained concepts present in each image. We use the label set from Koh et al. [6] which has 1:1 mapping with the ground-truth concepts in the CUB dataset. We use precision and recall metric to measure the quality of annotations from Grouding DINO for each concepts. Table D.1 present mean precision and mean recall value at different confidence threshold. We observe that the obtained annotations have a very high recall i.e if the concept is present in the image, grounding DINO is able to retrieve the object. The precision is also sufficiently high though it suffers from a relatively higher false-positive detection rate compared to false-negative detection rate. However, as demonstrated in our qualitative and quantitative studies (Table 3, Fig 4, K.2, K.1) the effect of false-positive is minimal and VLG-CBM is able to faithfully represent concepts in the Concept Bottleneck Layer.

| Confidence threshold | Mean Precision | Mean Recall |
|:---:|:---:|:---:|
| 0.10 | $0.7150 \pm 0.07$ | $0.9930 \pm 0.08$ |
| 0.15 | $0.7156 \pm 0.07$ | $0.9693 \pm 0.11$ |
| 0.20 | $0.7121 \pm 0.10$ | $0.8713 \pm 0.21$ |

Table D.1: Quantitative evaluation of concepts obtained from Grounding DINO model with Mean Precision and Recall for concepts at different confidence thresholds.

**VLG-CBM** provide accurate explanations while prior work provide inaccurate/wrong/less useful explanations!

## VLG-CBM (ours)

1. short pointed beak (0.65)
2. blue head (0.21)
3. green back (0.09)
4. short stout bill (0.01)
5. small songbird (0.01)
Sum of other concepts (**0.00**)👎

1. blue gray wings (9.40)
2. bright golden yellow plumage (1.31)
3. yellow head and breast (1.07)
4. large conical bill (0.53)
5. black mask on the face (0.26)
Sum of other concepts (0.00)👎

1. small forked tail (7.22)
2. black and white wings (3.57)
3. small delicate body (2.75)
4. long slender beak (0.12)
5. pointed beak (0.00)
Sum of other concepts (0.00)👎

## LF-CBM

1. NOT a brown and white color scheme (1.77)
2. NOT white and black coloration (1.66)
3. iridescent feathers (1.37)
4. NOT a black bib with white stripes (1.29)
5. NOT a black and white color scheme (1.01)
Sum of other concepts (4.22)

1. a yellow head (1.96)
2. NOT a red crest on the head (0.99)
3. orange legs (0.99)
4. yellow or orange plumage (0.80)
5. a bright orange breast (0.76)
Sum of other concepts (5.53)

1. NOT dark wingtips (2.07)
2. white breast with brown spots (1.93)
3. NOT a dark brown or black color (1.25)
4. yellow feet (1.12)
5. NOT long, blue-gray wings (0.81)
Sum of other concepts (5.12)

## LM4CV

1. Color - Blue head, olive back, yellow underparts (101.08)
2. grayish head, back, wings and tail with blue highlights (94.03)
3. bright blue and orange plumage (91.44)
4. large red bill with a slightly hooked tip (89.09)
5. distinctive white throat (-76.91)
Sum of other concepts (-34.89)

1. long, straight orange bill (141.48)
2. large, orange bill with a black tip (100.07)
3. pointed orange bill (95.72)
4. yellow and black plumage (84.85)
5. bright blue and orange plumage (76.97)
Sum of other concepts (-257.02)

1. red, black and white feathers (93.12)
2. bright red head and breast (84.05)
3. medium-sized black bird with bronze or brownish-gray highlights (-84.05)
4. dark gray wings with pale gray edging (-78.55)
5. bright red head and nape (75.39)
Sum of other concepts (113.93)

Figure D.1: Full version of Fig 1 comparing explanation of LF-CBM and LM4CV with VLG-CBM(ours)

## E    Distribution of nonzero weights among class

The NEC metric controls the average number of non-zero weights among classes. Further, we study the distribution of non-zero weight numbers between different classes. We choose our VLG-CBM model trained on CUB and places365 datasets, which have 200 and 365 classes, respectively, and plot the distribution of non-zero weights. Both models are trained to have NEC=5 The results are shown in Fig. E.1. The figure suggests most classes have non-zero weight numbers around 5, while a small number of classes utilize more concepts to make decisions.

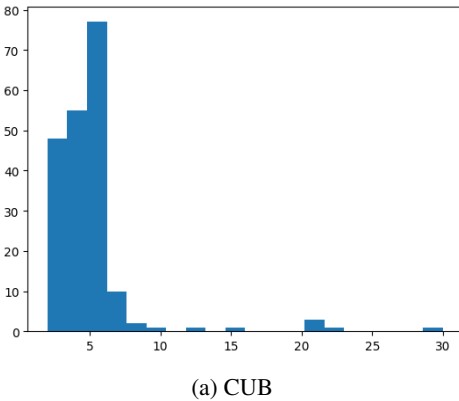
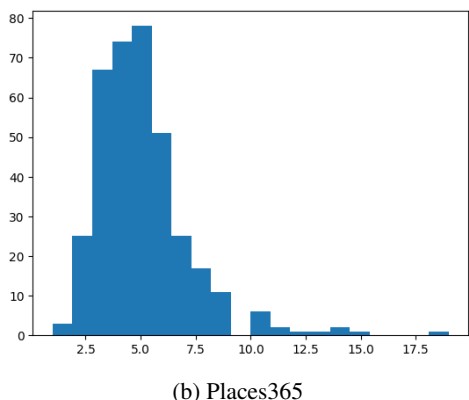

(a) CUB                                        (b) Places365

Figure E.1: distribution of non-zero weight numbers from CUB and Places365 dataset. The models are trained to have NEC=5.

## F    Constructing model with specified NEC

In this section, we discuss how to construct models with specified NEC. When using methods with dense final layers (e.g. [25]), controlling NEC is simply controlling total number of concepts in the concept set. Hence, below we mainly focus on models with sparse final layers.

When training the final linear layer, larger lambda(regularization strength) pushes the model to be sparser. Hence, we utilize GLM-SAGA[24], which allows us to obtain a regularization path consists of different lambdas. To be more specific, we choose a $\lambda_{max}$ and train models with $\lambda$ in $[\lambda_{min} = \lambda_{max}/500\lambda_{max}]$, and take 50 $\lambda$ evenly from the interval in log space. Then, we choose the weight matrix with the closest NEC and pruning the weights from smallest magnitude to largest to enforce strict NEC. Hence, the actual NEC is enforced to be exactly as prespecified ones.

## G    Additional case study examples

### G.1    Negative concepts in reasoning

In LF-CBM [15] and our VLG-CBM, normalization is applied on concept logits before the final decision layer. Hence, a negative value of concept logits indicates corresponding concept does not appear in the image. Following LF-CBM, we mark these concepts as "NOT {concept}" in explaining the decision. To study the frequency of this negative reasoning, we count the times these negative concepts appear in top-5 contributing concepts on CUB dataset. The results show that, for VLG-CBM, 162 out of 28950(0.56%) reasonings are through negative concepts. For comparison, LF-CBM utilizes 6687 out of 28950(23.10%) negative reasoning.

### G.2    Impact of NEC

The study in Section 5.3 shows that our VLG-CBM provides more interpretable decisions than baseline methods. To better understanding where these advantages comes from, we conduct a further study to set the baselines with NEC=5 and compare the decision interpretation, see Figs. G.2 to G.4. The results suggest setting NEC=5 alleviate the problem from non-top-5 concepts. However, wrong/inaccurate/less useful explanations still exist.

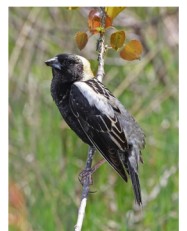

Ground truth: **Bobolink**.
VLG-CBM prediction: **Bobolink**

1. black v on the back(4.30)
2. black and white striped head(1.94)
3. black head and back(0.27)
4. small round body(0.07)
5. NOT white stripes above the eyes(0.01)
Sum of other concepts: (0.03)

Figure G.1: Image 307: An example of negative reasoning of VLG-CBM

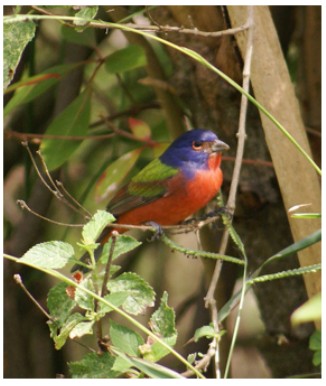

**LF-CBM**

1. NOT a brown and white color scheme (1.77)
2. NOT white and black coloration (1.66)
3. iridescent feathers (1.37)
4. NOT a black bib with white stripes (1.29)
5. NOT a black and white color scheme (1.01)
Sum of other concepts (4.22)

**LF-CBM (NEC=5)**

1. NOT a brown and white color scheme(2.39)
2. NOT a black bib with white stripes(1.08)
3. NOT a black and white color scheme(0.61)
4. iridescent feathers(0.27)
5. NOT yellowish-brown wings(0.25)
Sum of other concepts: (0.00)

**LM4CV**

1. Color - Blue head, olive back, yellow underparts (101.08)
2. grayish head, back, wings and tail with blue highlights (94.03)
3. bright blue and orange plumage (91.44)
4. large red bill with a slightly hooked tip (89.09)
5. white rump patch at the base of the tail (59.69)
Sum of other concepts (-171.50)

**LM4CV (NEC=5)**

1. bright reddish brown head, crown and back of neck.(382.61)
2. bright yellow, green and blue plumage(95.61)
3. bright yellow throat, breast, and flanks with black bars (51.36)
4. Broad tail that is shorter than other pelican species (-36.48)
5. Mottled brown on the nape, mantle, and scapulars(-243.90)
Sum of other concepts: (0.00)

**LaBo**

1. beautiful bird with a brightly colored body (0.02)
2. small, plump songbird with a short tail and a pointed bill (0.02)
3. beautiful bird with a brightly colored plumage (0.02)
4. one of the most beautiful north american songbirds (0.02)
5. colors are very vibrant and beautiful (0.02)
Sum of other concepts (41.25)

**LaBo(NEC=5)**

1. beautiful little bird with a very colorful plumage(3.98)
2. very colorful bird, with a lot of blue and green(0.43)
3. very pretty and very colorful(0.41)
4. NOT white stripes on white stripes on brown(0.22)
5. known as the "rainbow jay" due to its bright plumage(0.11)
Sum of other concepts: (0.00)

Figure G.2: Comparing baselines with different NECs

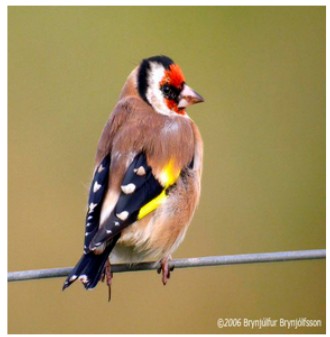

## LF-CBM

1. NOT dark wingtips(2.07)
2. white breast with brown spots(1.93)
3. NOT a dark brown or black color(1.25)
4. yellow feet(1.12)
5. NOT long, blue-gray wings(0.81)
Sum of other concepts: (5.12)

## LM4CV

1. red, black and white feathers(93.11)
2. bright red head and breast(84.05)
3. bright red head and nape(75.39)
4. bright red crescent below its beak (63.14)
5. White neck with a black collar and chestnut red head and breast(60.52)
Sum of other concepts: (-172.32)

## LaBo

1. male goldfinch is the more brightly colored of the sexes, with(0.02)
2. seen in flocks of other goldfinches(0.02)
3. often forming flocks with other goldfinches(0.02)
4. visit bird tables and feeders(0.02)
5. young goldfinches are drabber than adults, with brownish plumage(0.02)
Sum of other concepts: (41.69)

## LF-CBM (NEC=5)

1. NOT dark wingtips(1.80)
2. NOT long, blue-gray wings(0.69)
3. yellow feet(0.66)
4. a red face(0.08)
5. a Scarlet-red body(0.00)
Sum of other concepts: (0.00)

## LM4CV (NEC=5)

1. bright reddish brown head, crown and back of neck.(344.38)
2. bright yellow, green and blue plumage(87.41)
3. bright yellow throat, breast, and flanks with black bars (39.26)
4. Broad tail that is shorter than other pelican species (-34.77)
5. Mottled brown on the nape, mantle, and scapulars(-227.34)
Sum of other concepts: (0.00)

## LaBo(NEC=5)

1. closely related to the goldfinch(3.83)
2. often forming flocks with other goldfinches(1.16)
3. NOT from alaska and canada to the southwestern united states(1.10)
4. often forming flocks with other goldfinches and similar small birds(0.17)
5. NOT found in eastern and central united states year-round(0.03)
Sum of other concepts: (0.02)

Figure G.3: Comparing baselines with different NECs

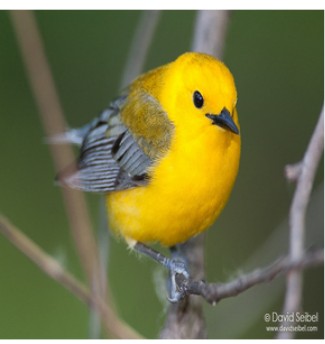

## LF-CBM

1. a yellow head(1.96)
2. NOT a red crest on the head(0.99)
3. orange legs(0.99)
4. yellow or orange plumage(0.80)
5. a bright orange breast(0.76)
Sum of other concepts: (5.53)

## LM4CV

1. long, straight orange bill (141.48)
2. large, orange bill with a black tip (100.07)
3. pointed orange bill (95.72)
4. yellow and black plumage(84.85)
5. bright blue and orange plumage(76.97)
Sum of other concepts: (-257.02)

## LaBo

1. is the only warbler with entirely(0.02)
2. largest warbler in north america(0.02)
3. plumage is bright yellow(0.02)
4. largest and heaviest member of the wood-warbler family(0.02)
5. yellow bird(0.02)
Sum of other concepts: (42.29)

## LF-CBM (NEC=5)

1. a yellow head(2.36)
2. yellow or orange plumage(0.60)
3. orange legs(0.17)
4. a black ring around the bill(0.08)
5. Glossy black wings(0.00)
Sum of other concepts: (0.00)

## LM4CV (NEC=5)

1. bright yellow throat, breast, and flanks with black bars (320.88)
2. bright yellow, green and blue plumage(285.24)
3. bright reddish brown head, crown and back of neck.(-89.49)
4. Broad tail that is shorter than other pelican species (-126.68)
5. Mottled brown on the nape, mantle, and scapulars(-151.65)
Sum of other concepts: (0.00)

## LaBo(NEC=5)

1. NOT sometimes called the "sea sparrow" due to its black and white plumage(1.10)
2. yellow head, chest, and belly(0.95)
3. yellow head is thought to be a sign of maturity and wisdom(0.37)
4. orange in color(0.28)
5. series of high, thin "peeps".(0.15)
Sum of other concepts: (0.37)

Figure G.4: Comparing baselines with different NECs

# H  Further discussion on decision explanations

In this section, we further discuss some interesting phenomena observed in the decision explanations generated by different models.

## H.1  Negative contributions

In Fig. 1, we could see that LM4CV[25] generates negative contribution values, while LF-CBM[15] and our VLG-CBM do not. We hypothesize the reason is different training methods: LM4CV trains a dense final layer, hence the concepts irrelevant to the class may provide a negative contribution. LF-CBM and VLG-CBM, however, train a sparse final layer. To enforce sparsity, the model only captures relevant concepts for decision. Hence, it's natural to expect most contributions should be positive.

# I  Additional experiment results

## I.1  Generalizability to OOD datasets

In this section, we study a question: will our VLG-CBM hurts the generalization ability of original model to Out-Of-Distribution(OOD) dataset? To study this problem, we conduct experiment on Waterbirds dataset [19]. Waterbirds is an OOD dataset adapted from the CUB dataset, which combines bird photos from CUB with image backgrounds from Places365. We use the same ResNet model as we used in Table 3 for CUB dataset. For VLG-CBM, we choose NEC=5 and compare the results with the standard, non-interpretable models. On this dataset, the results are shown below: It can be seen

| Method | CUB Accuracy | Waterbirds Accuracy |
|---|---|---|
| Standard model (black-box) | 76.70% | 69.83% |
| VLG-CBM | 75.79% | 69.83% |

Table I.1: Accuracy of VLG-CBM and standard blackbox model on CUB and Waterbirds datasets.

that our VLG-CBM generalizes well as the standard model does, which shows that our VLG-CBM is competitive and has very small accuracy trade-off with the interpretability compared with the standard black-box model.

## I.2  Ablation study

### I.2.1  Ablation on augmentation probability

In this section, we conduct an ablation study on the probability of applying our crop-to-concept data augmentation introduced in Section 3.1. The dataset we used is the CUB dataset and the backbone is ResNet as we used in the main experiment. The results are listed below. From the table, we could see

| Crop-to-Concept-Prob | ANEC-5 | ANEC-avg |
|---|---|---|
| 0.0 | 75.73 | 75.76 |
| 0.2 | **75.83** | **75.88** |
| 0.4 | 75.71 | 75.72 |
| 0.6 | 75.57 | 75.62 |
| 0.8 | 75.52 | 75.57 |
| 1.0 | 72.29 | 73.15 |

that the performance is best with augmentation probability 0.2.

# J  Human study

In this section, we present a human study following the practice of Oikarinen et al. [15] on Amazon MTurk platform. To briefly summarize, we show the annotator top-5 contributing concepts of our method (VLG-CBM) and baseline (LF-CBM or LM4CV) and asking them which one is better.

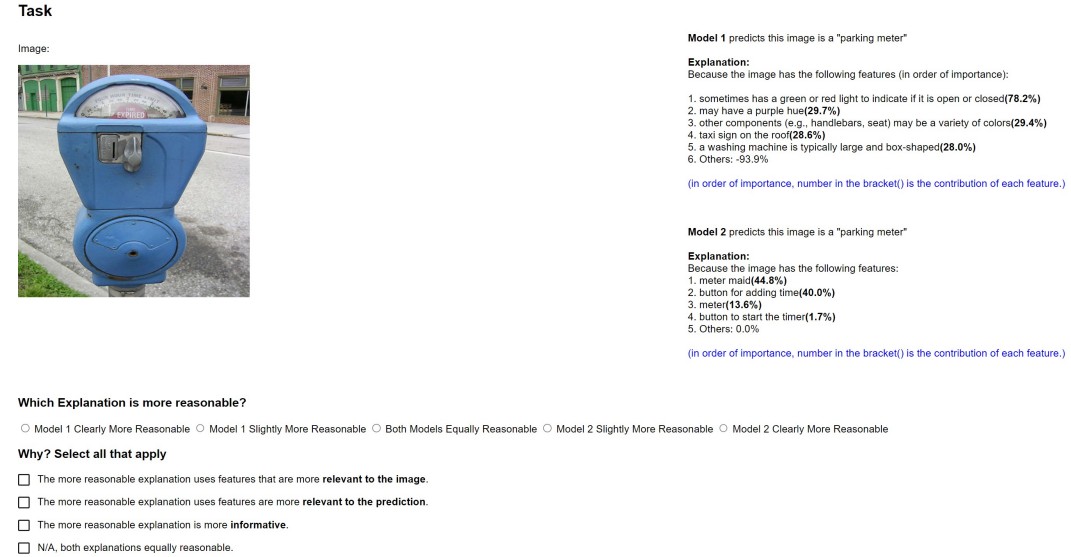

Figure J.1: An example of human study interface

The scores for each method are assigned as 1-5 according to the response of annotators: 5 for the explanations from VLG-CBM is strongly more reasonable, 4 for VLG-CBM is slightly more reasonable, 3 for both models are equally reasonable, 2 for the baseline is slightly more reasonable, and 1 for the baseline is strongly more reasonable. Thus, if our model provides better explanations than the baselines, then we should see a score higher than 3. We show an example screenshot of our study in Fig. J.1.

We report the average score in Table J.1 for two baselines: LF-CBM and LM4CV. For each baseline, we randomly sample 200 images and collect 3 results from 3 different annotators. It can be seen that VLG-CBM has scores higher than 3 for both baselines, indicating our VLG-CBM provides better explanations than both baselines. LaBo is excluded in our experiment due to its dense layer and large number of concepts: the top-5 concepts usually account for less than 0.01% of final prediction.

| Experiment | Score (VLG-CBM) | Score (Baseline) |
|---|---|---|
| VLG-CBM vs. LF-CBM | 3.33 (1.54) | 2.67 (1.54) |
| VLG-CBM vs. LM4CV | 3.38 (1.54) | **2.62 (1.54)** |

Table J.1: Average Mturk score for our VLG-CBM and two baselines.

# K    Visualizing VLG-CBM explanations

This section presents an extended version of Table 4 visualizing top-5 images for randomly picked concepts for CUB and Places365 dataset. The results are shown in Fig K.1, K.3, K.4, and K.5 for Places365 and K.2, Fig K.6, Fig K.7, and Fig K.8 for CUB.

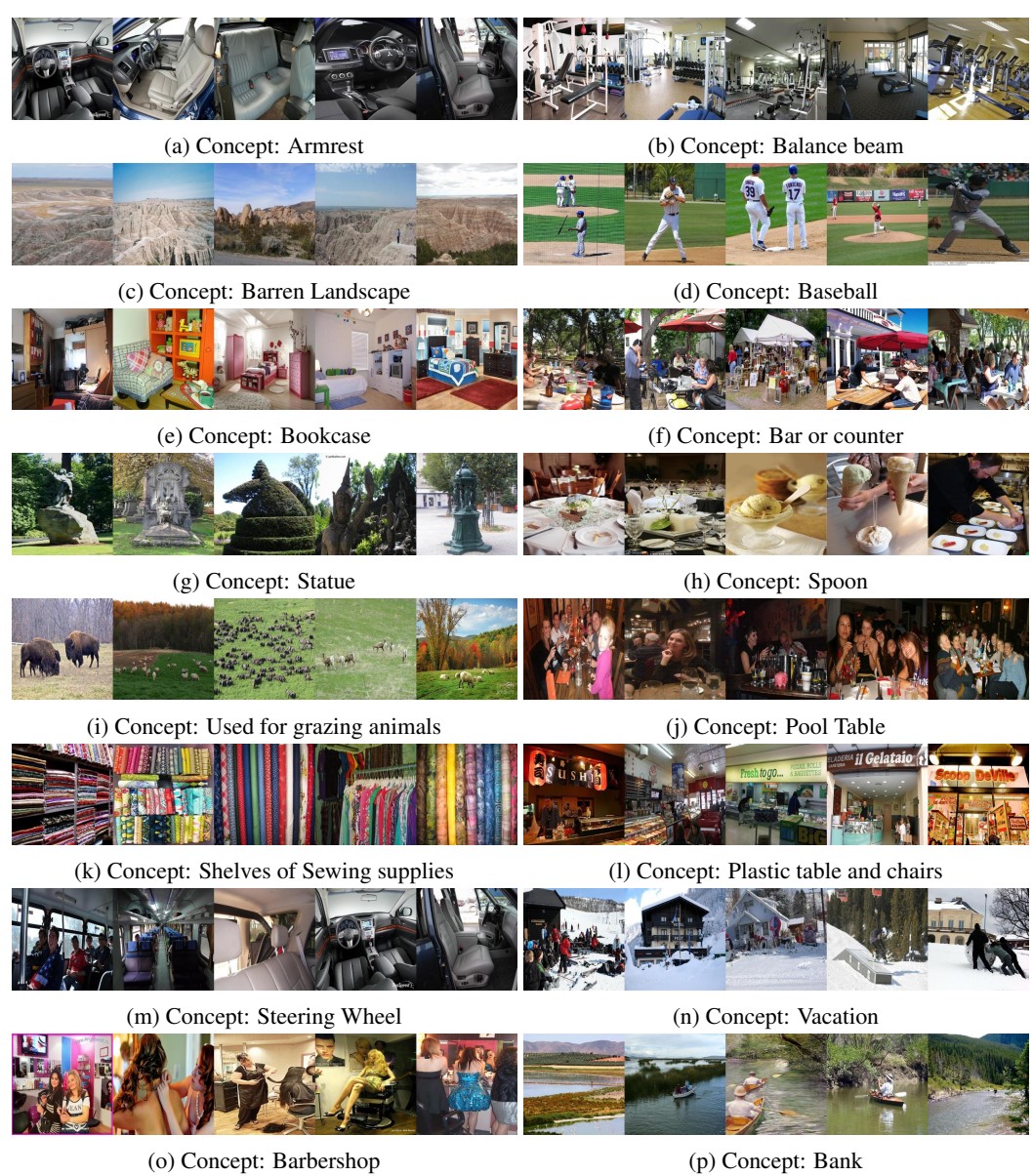

(a) Concept: Armrest        (b) Concept: Balance beam

(c) Concept: Barren Landscape        (d) Concept: Baseball

(e) Concept: Bookcase        (f) Concept: Bar or counter

(g) Concept: Statue        (h) Concept: Spoon

(i) Concept: Used for grazing animals        (j) Concept: Pool Table

(k) Concept: Shelves of Sewing supplies        (l) Concept: Plastic table and chairs

(m) Concept: Steering Wheel        (n) Concept: Vacation

(o) Concept: Barbershop        (p) Concept: Bank

Figure K.1: Top-5 activating images for randomly selected Places365 concepts

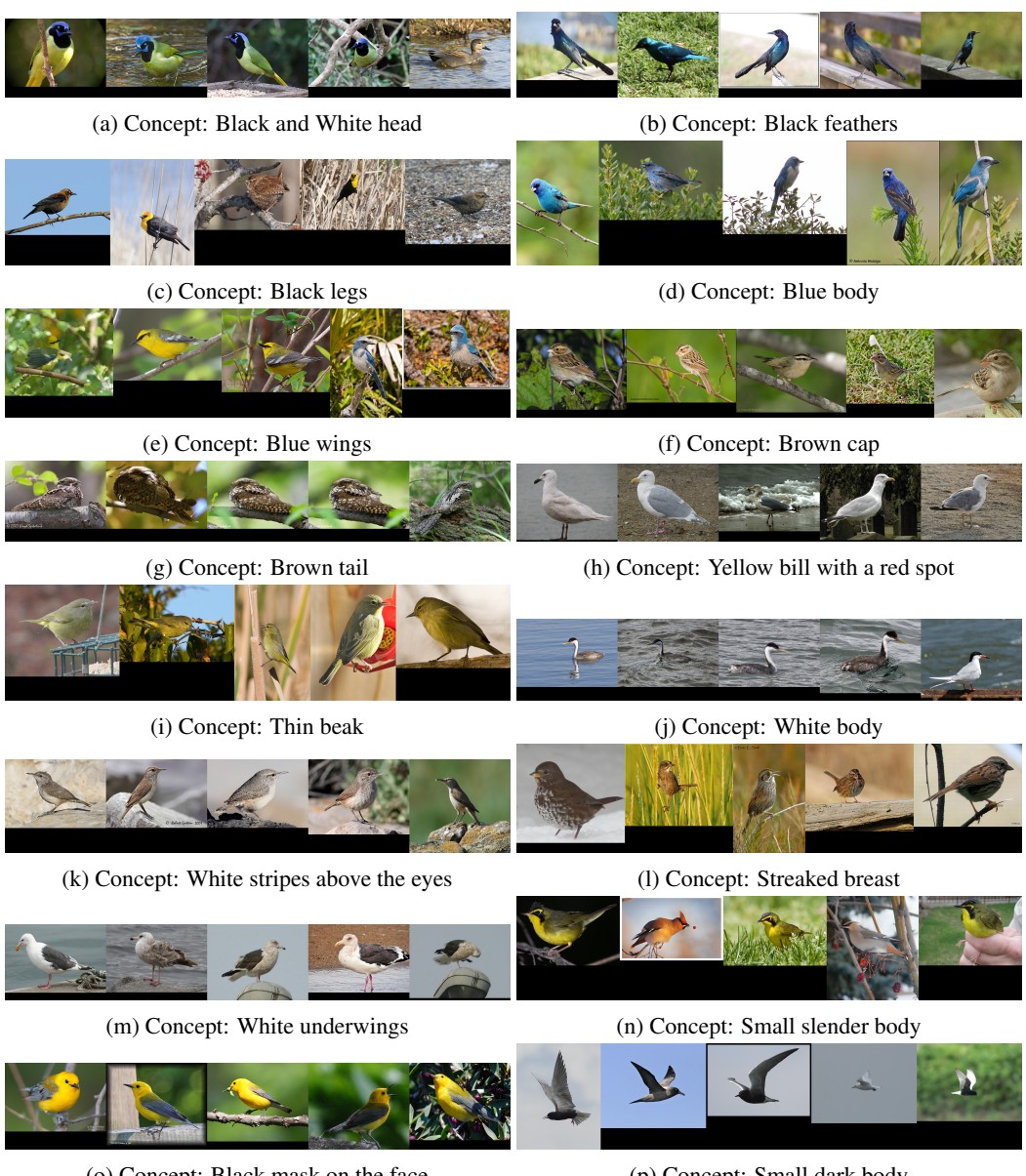

(a) Concept: Black and White head

(b) Concept: Black feathers

(c) Concept: Black legs

(d) Concept: Blue body

(e) Concept: Blue wings

(f) Concept: Brown cap

(g) Concept: Brown tail

(h) Concept: Yellow bill with a red spot

(i) Concept: Thin beak

(j) Concept: White body

(k) Concept: White stripes above the eyes

(l) Concept: Streaked breast

(m) Concept: White underwings

(n) Concept: Small slender body

(o) Concept: Black mask on the face

(p) Concept: Small dark body

Figure K.2: Top-5 activating images for randomly selected CUB concepts

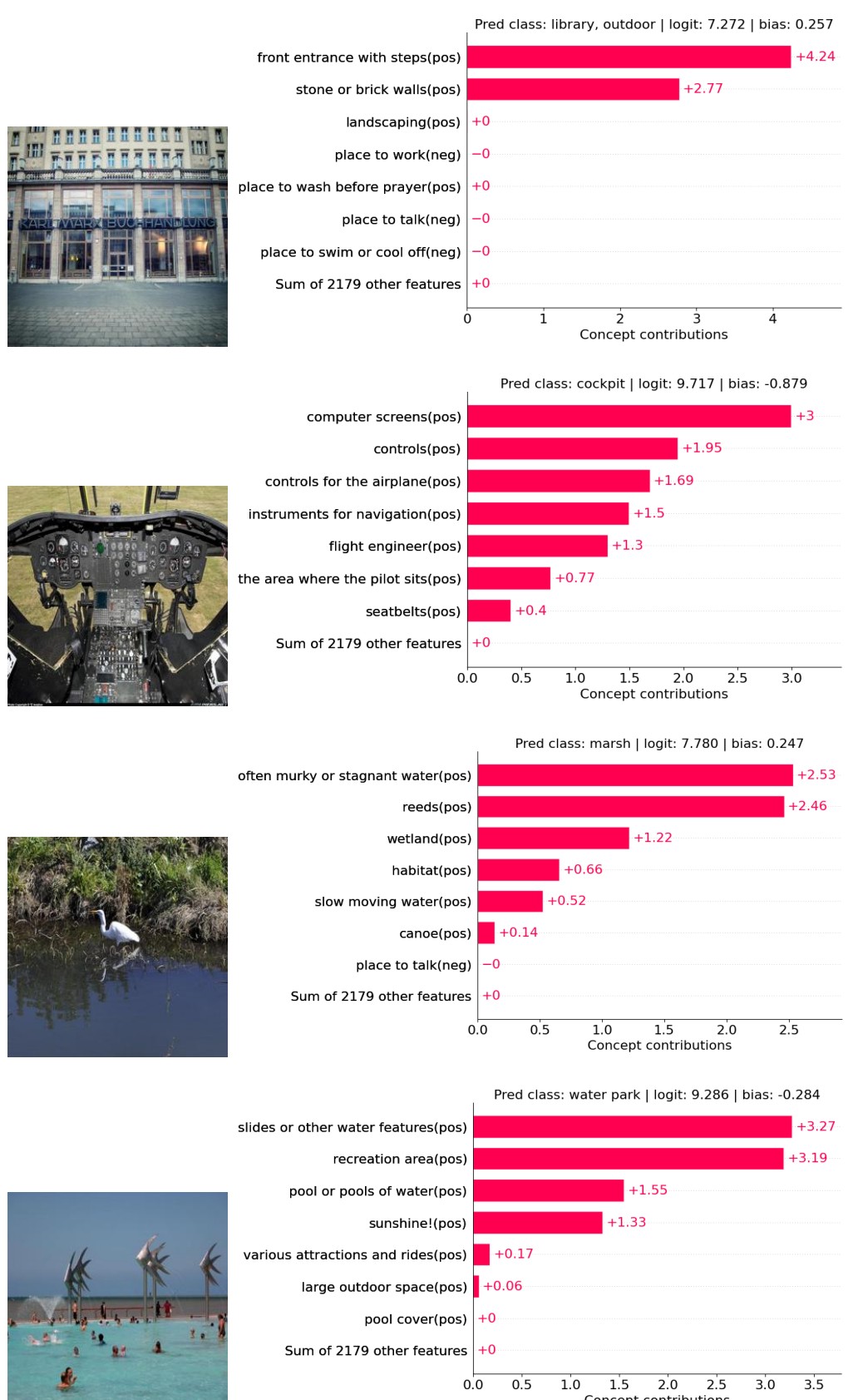

Figure K.3: Randomly selected explanations for Places365 (Part 1)

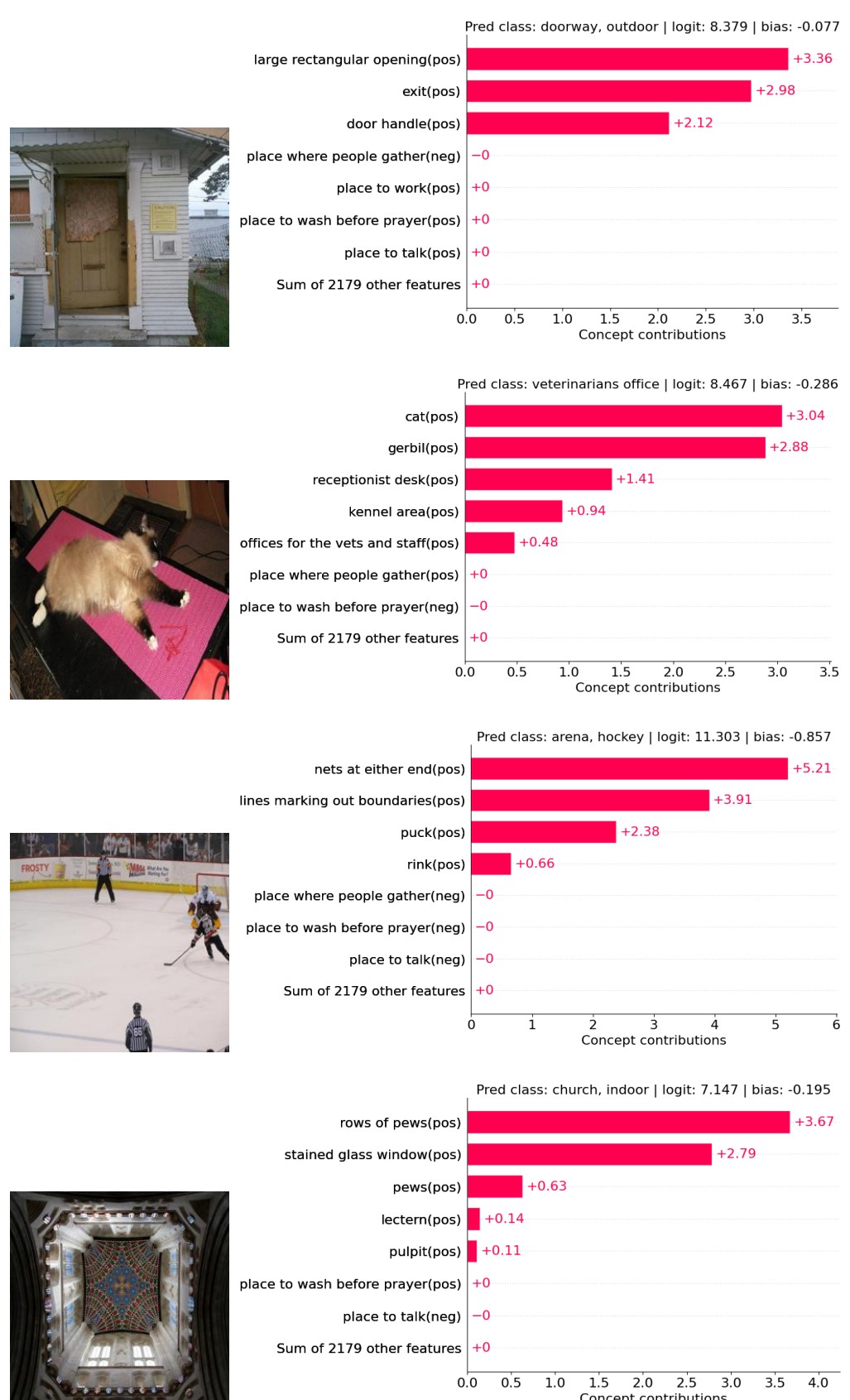

Figure K.4: Randomly selected explanations for Places365 (Part 2)

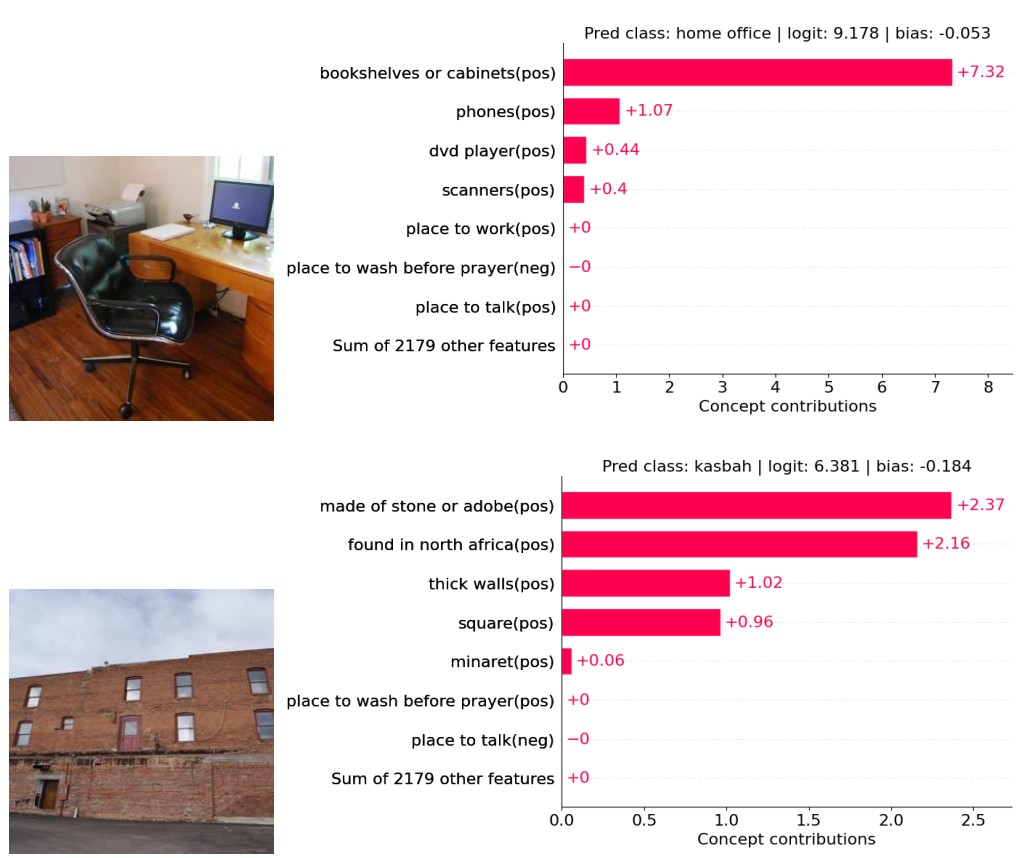

Figure K.5: Randomly selected explanations for Places365 (Part 3)

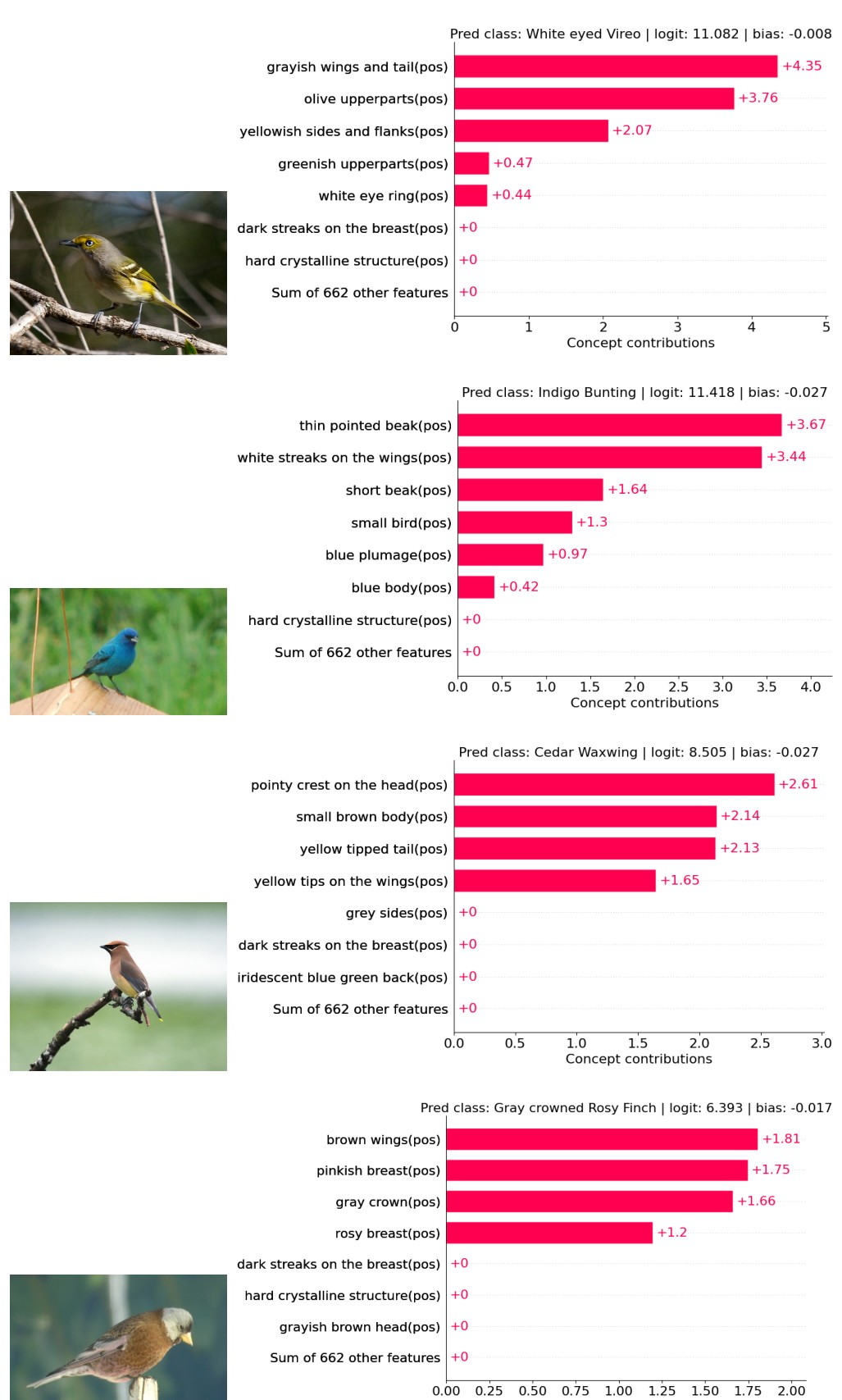

Figure K.6: Randomly selected explanations for CUB (Part 1)

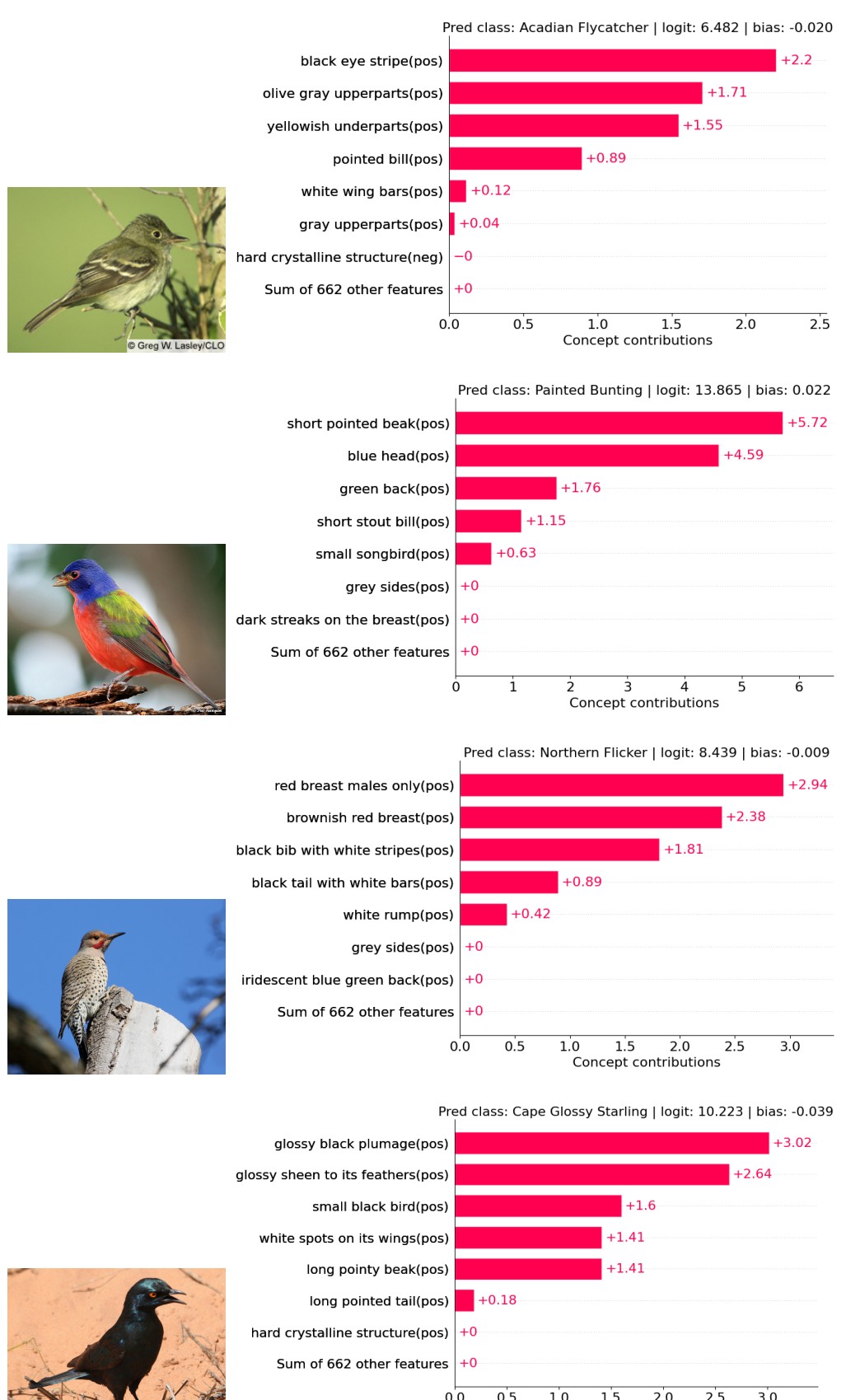

Figure K.7: Randomly selected explanations for CUB (Part 2)

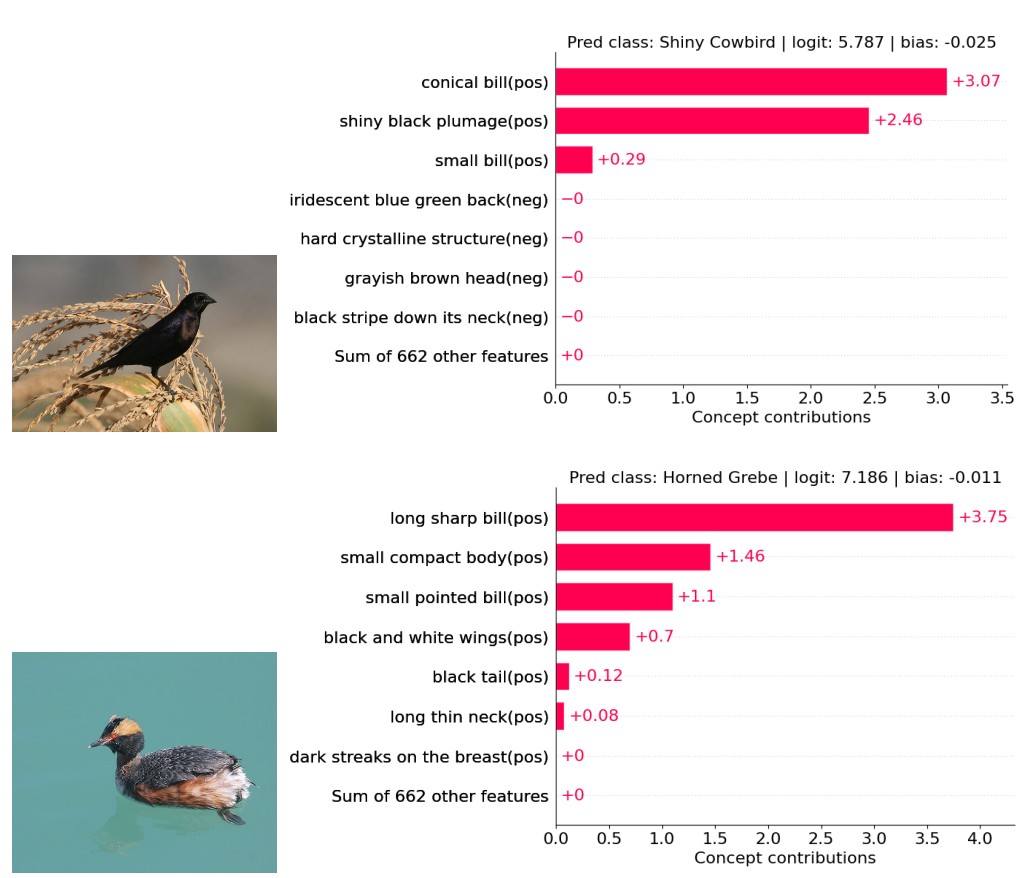

Figure K.8: Randomly selected explanations for CUB (Part 3)

