# OpenReview forum: "VLG-CBM: Training Concept Bottleneck Models with Vision-Language Guidance"
_NeurIPS.cc/2024/Conference — NeurIPS 2024 poster_

### Official Review · Reviewer_i5YZ · 2024-07-13

**Soundness:** 3
**Presentation:** 3
**Contribution:** 2
**Rating:** 5
**Confidence:** 4

**Summary:**

In this paper, the authors propose a learning pipeline named VLG-CBM, which enhances CBM with the aid of open-domain object detectors to create more accurate concept labels, while also excluding image-irrelevant concepts (e.g,. "loud music").
Also, in order to prevent the 'information leak' in CBMs, which stands for a phenomenon where the exploitation of irrelevant concepts may help downstream accuracy, authors propose a metric named Number of Effective Concepts (NEC) which enables the fair comparison of different CBMs within a similar amount of information being provided.
Experimental results show that with the proposed VLM-CBM, high-quality concept labels are obtained therefore achieving better performances within similar NEC.

**Strengths:**

1. The idea of applying a grounding model to generate an accurate concept label is sound and seems effective as shown in the experimental results.

2. The theoretical analysis of random CBM well explains the problem of information leaks in CBMs, and the proposed NEC is a proper method to measure the information being provided to a CBM classifier, enabling fair comparison of CBMs within a similar amount of information being provided.

3. The paper is generally well-written, making it easy to follow the main ideas and claims.

**Weaknesses:**

1. Only experimental results under controlled NEC are reported, making it difficult to compare the performance of VLG-CBMs with previous works in conventional settings. For instance, the average accuracy of LaBo in the CIFAR-100 dataset under an NEC-controlled setting is reported as 55.18%, while the full performance in the original paper is around 86% (86.82 in the dev set, 86.04 in the test set). Therefore, also comparing the performance of VLG-CBM under conventional settings in previous work would make the comparison easier.

2. The idea of applying a detection or a grounding module to generate an image annotation is already widely explored (e.g., KOSMOS-2 [1]). The proposed data generation process does not significantly differ from previous approaches, limiting the originality of the proposed method.


[1] Peng et al., KOSMOS-2: Grounding Multimodal Large Language Models to the World.

**Questions:**

1. Questions regarding Table 3. 1) How is the performance of each method? Only the percentage of changed decisions are reported, therefore providing accuracies may help a better understanding of the effectiveness of each method. 2) Isn't it obvious that the percentage of changed decisions is lower in VLG-CBM since the sparsity constraint is not applied to other methods? Can you also provide the same analysis with sparsity constraint applied to other methods?

2. Providing qualitative examples of labeling results using Grounding-DINO would help in understanding the quality of the auto-labeled dataset, which seems to be absent in the current version of the paper.

3. Also, please address the concerns elaborated in the weakness section.

**Limitations:**

Authors have described limitations and possible societal impacts in the main paper and the supplementary material.

---

> ### Author Rebuttal · Authors · 2024-08-07
>
> Dear Reviewer i5YZ,
>
> Thank you for your feedback. Below are responses to your comments.
>
> **Q1:** Therefore, also comparing the performance of VLG-CBM under conventional settings in previous work would make the comparison easier.
>
> **A1:** In this paper we do not compare the results under a dense final layer setting mainly due to this setting is less informative: as shown in Fig. 3 of our draft, even random baseline achieve similar performance to SOTA CBM models, when NEC becomes large enough.
>
> ---
>
> **Q2:** The idea of applying a detection or a grounding module to generate an image annotation is already widely explored (e.g., KOSMOS-2 [1]). The proposed data generation process does not significantly differ from previous approaches, limiting the originality of the proposed method.
>
> **A2:** We would like to clarify our contribution in regards to grounding concepts for training CBMs. CBMs rely on LLMs for a concept set which usually provide generic concepts and might not be possible to visualize in images. We are the first to use open-vocabulary grounding models to improve the effectiveness of CBMs.
>
> ---
>
> **Q3:** How is the performance of each method? Only the percentage of changed decisions are reported, therefore providing accuracies may help a better understanding of the effectiveness of each method
>
> **A3:** We calculate the performance of each method:
>
>
> |        | Full   | Truncated |
> |--------|--------|-----------|
> | LF-CBM | 73.92% |    44.80% |
> | LM4CV  | 43.36% |     0.01% |
> | LaBo   | 81.70% |    17.53% |
>
> We could see here most methods suffer significant performance drops. LF-CBM has a smaller gap, partially due to its sparse final layer.
>
> ---
>
> **Q4:** Isn't it obvious that the percentage of changed decisions is lower in VLG-CBM since the sparsity constraint is not applied to other methods? Can you also provide the same analysis with sparsity constraint applied to other methods?
>
> **A4:** Yes, what we want to illustrate here is to emphasize the importance of NEC control. A quite popular method to explain model decisions for CBMs is to show the users the top-5 contributing concepts. However, we found this approach has potential risk without NEC control: as shown in Tab. 3, if limited to only top-5 concepts, the decision of the model will change significantly. This suggests that showing top-5 concepts does not faithfully explain the model behavior.
>
> To better illustrate this, we choose a baseline, LF-CBM and apply our NEC control on it. The below table shows that controlling NEC effectively reduces the decision change.
>
> | NEC                         | Change percentage |
> |-----------------------------|-------------------|
> |                           5 |            15.77% |
> |                          10 |            35.39% |
> |                          15 |            37.70% |
> | original LF-CBM (NEC=31.52) |            40.79% |
>
> ---
>
> **Q5:** Providing qualitative examples of labeling results using Grounding-DINO would help in understanding the quality of the auto-labeled dataset, which seems to be absent in the current version of the paper.
>
> **A5:** We have provided few examples of annotation obtained from GroundingDINO in **global response supplementary pdf**.
>
> ---
>
> **Summary**
>
> In summary, we have
>
> * In **Q1**, we discuss why conventional benchmark is not that informative.
>
> * In **Q2**, we clarify our difference with previous work.
>
> * In **Q3**, we provide the performance change of each method in Table. 3.
>
> * In **Q4**, we discuss how NEC constraint affect the decision changes and add LF-CBM experiments to support it.
>
> * In **Q5**, we provide a few examples from GroundingDINO.
>
> In response to the weakness part, we have
>
> * addressed the reviewer’s concern in weakness #1 in **Q1**.
>
> * addressed the reviewer’s concern in weakness #2 by clarifying our difference with previous work in **Q2**.
>
>
> We believe that we have addressed all your concerns. Please let us know if you still have any reservations and we would be happy to address them!

---

> ### Comment · Reviewer_i5YZ · 2024-08-12
>
> Thank you for your response in a limited time.
>
> Although some of my concerns have been addressed (e.g., questions regarding Table 3 and request for qualitative results), I still feel that the contribution is somewhat limited, which is also pointed out by reviewer QueL.
>
> For me, the novel point of this paper seems to be
> 1. Introducing the idea of utilizing Grounding-DINO to remove noisy concepts generated by LLMs,
> 2. Proposing the new evaluation metric, NEC, quantifying the previously defined problem of information leaks.
>
> Since the problem of non-visual concepts being generated by LLMs [1] and information leaks in CBM [2, 3] have been discovered before, it seems that the novelty in the problem definition or its solution is somewhat limited for me.
>
> Therefore, could the authors provide additional points that I am missing that make the problem definition or its solution's novelty clearer?
>
> [1] Kim et al., Concept bottleneck with visual concept filtering for explainable medical image classification, MICCAI Workshop 2023
>
> [2] Roth et al., Waffling around for Performance: Visual Classification with Random Words and Broad Concepts, ICCV 2023
>
> [3] Yan et al., Learning Concise and Descriptive Attributes for Visual Recognition, ICCV 2023

---

> > ### Author Response · Authors · 2024-08-13
> > **Clarification on our novelty and contributions - Part 1**
> >
> > Dear Reviewer i5YZ,
> >
> > Thanks for your response! We are happy to learn that some of your concerns have been addressed in our rebuttal response, and we appreciate the opportunity to provide you with additional clarifications regarding our contributions and the novelty of our methods as requested in the new comments.
> >
> > **#1 Our main contributions and novelty**
> >
> > We would like to clarify that our contribution is not limited to the two as the reviewer has mentioned, and the Reviewer QueL actually has some misunderstandings of our contributions which we clarified in the new response (please see post title: **Clarification on misunderstanding of our contributions - Part 1 & Clarification on misunderstanding of our contributions - Part 2**) and our original rebuttal response to Reviewer QueL (please see our response **A1 and A2**).
> >
> > Below we provide the details of our contributions, explain why our work is novel, and compare with the reference [1-3] as provided by the reviewer.
> >
> > First of all, there are **three** main contributions of our work:
> > * **Contribution (1):** the **first** end-to-end pipeline to build CBM with vision guidance from open-vocabulary object detectors in Sec 3;
> > * **Contribution (2):** the **first** rigorous theoretical analysis to prove that CBMs have serious issues on information leakage in Sec 4.1, whereas existing work [3, 4] only have empirical observations;
> > * **Contribution (3):** a **new and useful** metric to control information leakage problems that can facilitate fair comparison between different CBMs in Sec 4.2 and Sec 5.
> >
> >
> > Our **Contribution (1) and (3)** has been somehow (but not fully) recognized by the reviewer and our **Contribution (2)** was missed by the reviewer. The reviewer is correct that our **Contribution (1)** is to address non-visual concepts generated by LLMs and our **Contribution (3)** is to address the information leaks in CBM. Although there is some previous work [1-3] mentioned by the reviewer, they are limited by the following points:
> > * These works do not solve the problem fully
> >   * e.g. [1] is more related to Labo and still uses a CLIP-backbone, whereas our pipeline leverages open-vocabulary object detectors and is more comprehensive and can provide localized concepts information. Please see our below response **#2** for more details.
> >   * e.g. [3] only has empirical observation and did not provide theoretical analysis, whereas we provide the first theoretical analysis and proof to analyze the random concepts in CBMs. Please see our below response **#3 and #4** for more details.
> >
> > * Some of the work is not solving the same problem as we did
> >   * e.g. [2] focus on CLIP zero-shot classification, which is a different problem from what we are solving for random concepts in the CBMs in Sec 4. Please see our  below response **#4** for more details.
> >
> > Together with our 3 contributions, the extensive evaluations in sec 5 (see Table 2, 3, Figure 1,3, 4), and global response in the rebuttal (Table R1, R2, additional experiments, ablation study and human study) supports the superiority of our method and shows that our VLG-CBM framework is a promising approach towards building a faithful, reliable, and high performance CBM. More specifically, our method provides more favorable explanations than the baselines by around 18% in human study, our method also achieves up to 29.78% improvement on average accuracy and gets up to 51.09% improvement on accuracy at NEC=5.
> >
> > ---
> >
> > #2 Details of our **Contribution (1)** and how is our work different from [1]?
> >
> > To our best knowledge, there is no other work in the CBM literature that has the idea of leveraging object detection to address the concept faithfulness problem in CBMs. We are the first to provide this elegant and effective approach to address the inaccurate concept prediction problems. In [1], the key idea is to filter out the non-visual concepts by adding a vision activation term to the concept selection step. However, the main training pipeline is still CLIP-based, similar to LaBo. Additionally, our method in Sec 3 is more comprehensive and utilizes open-vocabulary object detectors in multiple stages of CBM pipeline: Obtaining annotations for multi-class training, filtering non-visual concepts, and utilizing predicted bounding boxes for improving classification performance. We will cite the paper [1] and discuss the above differences in the revised manuscript.

---

> > > ### Author Response · Authors · 2024-08-13
> > > **Clarification on our novelty and contributions - Part 2**
> > >
> > > Additionally, we would like to clarify that Grounding-DINO is not applied to just removing the concept set. In fact, removing noisy LLM-based concepts is just a natural outcome of our method, and this idea is actually more powerful than that: Based on Grounding-DINO we design a novel training pipeline: unlike CLIP, the Grounding-DINO model could generate bounding boxes for each concept. Our VLG-CBM designed a data augmentation mechanism (line 134-136 of the draft) to utilize this spatial and localized information, which is a key novelty compared to previous CLIP-based methods like [1].
> > >
> > > ---
> > >
> > > #3 Details of our **Contribution (2)**
> > >
> > > This is a contribution that has been missed by the reviewer and we would like to give more details on why this contribution is novel and important. As described in #1, our **Contribution (2)** is the **first** rigorous theoretical analysis to prove that CBMs have serious issues on information leakage in Sec 4.1.
> > >
> > >
> > > As we pointed out in the original manuscript (Line 40-42, Line 152-162), information leakage is an important issue that was observed by prior work [3, 4]. However, there is no rigorous theoretical understanding of this phenomenon in [3, 4], i.e. why CBM can achieve good accuracy even on random concepts? In fact, we are the first work to provide the first theoretical analysis and proof to explain this phenomenon: we prove in Thm 4.1 that a CBM can achieve optimal accuracy even on random conconcepts when the number of concepts is large enough.
> > >
> > > Our Thm 4.1 suggests that the number of concepts is an important factor of information leakage, and it also show that how the approximation error is controlled by number of concepts, which inspired us to control information leakage via the novel NEC metric that we propose in Sec 4.2 (which is our **Contribution (3)**).
> > >
> > > ---
> > >
> > > #4 Details of our **Contribution (3)** and how is our work different from [2, 3]?
> > >
> > > As we clarified in the above point **#1**, our **Contribution (3)** is a **new and useful** metric to control information leakage problems that can facilitate fair comparison between different CBMs in Sec 4.2 and Sec 5.
> > >
> > > As we described in the introduction of the draft (see lines 34-42), existing CBMs have serious problems of faithfulness due to information leakage problems. Without this metric, the performance of CBMs do not correctly reflect their “true” performance, which causes a false sense of confidence. Indeed, as we show in Table 2 in the draft and Table R1 in the Global response in the rebuttal, under the fair comparison setting, many existing CBMs' true performance are much worse than what they were reported. For example, using Acc@NEC=5, LF-CBM has around 2%-21% drop, Labo has around 20%-58% accuracy drop, LM4CV has around 34%-63% accuracy drop than reported, providing a false sense of performance.
> > >
> > > Additionally, we would like to highlight that our proposed new metric NEC is a theoretically grounded solution to control information leakage problems and has much more flexibility in concept choices and better interpretability compared with the previous approach [3]. In [3], the authors proposed to control the total number of concepts, which however limits the choice of concepts and can harm the interpretability due to dense connection, as supported by the human study results in **General Response #3** and the qualitative example in **Fig. 1** and **Fig. H.2 - H.4** in the appendix of our draft.
> > >
> > > For [2], we would like to clarify that [2] is mainly focused on using random words to improve the CLIP zero-shot classification, which is a different problem from what we are solving for random concepts in the CBMs in Sec 4. Although the results in [2] are interesting and show that using random words and characters could achieve comparable zero-shot classification performances, it is technically different from the information leakage problem that we study in CBMs in Sec 4.
> > >
> > > We will add the discussion of paper [2] in the revised draft and we have already cited [3] in our draft (please see line 155-158).
> > >
> > > ---
> > >
> > > ## Summary
> > > To sum up, we clarify that there are 3 main contributions in our work, clarify the novelty of our methods, and compare them to [1-3]. Specifically, we address the reviewer’s additional comments by
> > > * Clarifying our 3 contributions in **#1**
> > > * Clarifying the novelty of our **Contribution (1)** and compared it with **[1]** in **#2**
> > > * Clarifying the importance and novelty of our **Contribution (2)** in **#3**
> > > * Clarifying  the novelty of our **Contribution (3)** and compared it with **[2, 3]**  in **#4**
> > >
> > >
> > > Please let us know if you still have any concerns, we would be happy to discuss further.

---

> > > > ### Author Response · Authors · 2024-08-13
> > > > **Clarification on our novelty and contributions - Part 3**
> > > >
> > > > ## Reference
> > > > [1] Kim et al., Concept bottleneck with visual concept filtering for explainable medical image classification, MICCAI Workshop 2023
> > > >
> > > > [2] Roth et al., Waffling around for Performance: Visual Classification with Random Words and Broad Concepts, ICCV 2023
> > > >
> > > > [3] An et al. "Learning concise and descriptive attributes for visual recognition." ICCV 23
> > > >
> > > > [4] Mahinpei et al. “Promises and pitfalls of black-box concept learning models.” ICML 21

---

> ### Comment · Reviewer_i5YZ · 2024-08-14
>
> Thanks for your response.
> I appreciate the authors for providing a detailed comparison with previous works, which makes the position of the paper more clear.
>
> Since now I better understand the novelty of the paper, I am raising the score by one.
>
> Still, I highly recommend the authors include a detailed comparison with previous works in the manuscript.
> Since the contributions of the paper are built upon previous observations, clearly stating the novel points and importance of the Grounding-DINO approach and NEC metric may help readers better grasp the value of the paper.

---

> > ### Author Response · Authors · 2024-08-14
> > **Thank you for raising the score!**
> >
> > Dear Reviewer i5YZ,
> >
> > We are happy to see our clarification has addressed your concern on the contribution of our paper, and we appreciate your decision to raise the score to 5!
> >
> > We will follow your suggestion to update the manuscript with a detailed comparison and discussion with previous works and make it more clear about the novelty of our methods in the revised version.
> >
> > Thank you for the discussion and we appreciate your time and feedback!

---

### Official Review · Reviewer_MvHs · 2024-07-13

**Soundness:** 3
**Presentation:** 4
**Contribution:** 3
**Rating:** 7
**Confidence:** 4

**Summary:**

The paper uses foundational models to automate the generation of concept annotations which are then used to train a concept bottleneck model (CBM). To generate the concept set, the paper extends prior scalable CBM approaches[1] by using open-domain object detectors to weed out concepts that are not visually grounded. Concepts that are not present in any bounding-box for any training image are removed. Following existing work,  a CBM is trained sequentially – first, the concept layer is trained and frozen, then a sparse linear layer is trained. The paper introduces a theoretically justified metric, Number of effective concepts (NEC), which is the average number of concepts the sparse final layer uses to make the class prediction. NEC can be used to compare between various CBMs. Results are shown on 5 standard datasets, and they compare against 3 scalable CBM baselines.

[1] Oikarinen, Tuomas P. et al. “Label-Free Concept Bottleneck Models.” ArXiv abs/2304.06129 (2023): n. Pag.

**Strengths:**

\+ The paper is written well and easy to follow.
\+ The bounding boxes generated from open-domain object detectors are used to i) filter out concepts which are not visually grounded ii) augment training of the concept layer by cropping images to bounding-boxes
Both these design decisions seem crucial for learning a better, more interpretable concept layer.  I would find it interesting if this was studied in greater depth.
\+ The paper quantifies existing thought that having a dense prediction layer limits the applicability of CBMs. NEC can be used to fairly compare across CBMs.
\+ Empirical results are convincing. Their CBM outperforms existing baselines on classification accuracy across various levels of final layer sparsity.

**Weaknesses:**

\- It is not clear if NEC controls information leakage. As defined in [2], information leakage happens because of a `soft’ concept layer (that predicts probability of a concept instead of presence and absence). Though a small NEC reduces the possibility of information leakage and is a useful metric to check the goodness of the concept set.
\- A study of time and compute complexity is needed, as every image has to be passed through a large object-detector. The effort to generate concepts is much higher (compared to [1], who generate concepts class wise).

[1] Oikarinen, Tuomas P. et al. “Label-Free Concept Bottleneck Models.” ArXiv abs/2304.06129 (2023): n. Pag.
[2] Mahinpei, Anita et al. “Promises and Pitfalls of Black-Box Concept Learning Models.” ArXiv abs/2106.13314 (2021): n. pag.

**Questions:**

1. A study of the impact of proposed visually-grounded concept filtering would be interesting. For ex. What fraction of quantitative gains are because of the removed concept set? How much does augmentation contribute to overall performance gains?
2. As mentioned in the weaknesses section, a section on time complexity of concept set generation would be useful.
3. The theoretical analysis, though appreciated, seems out of place. The result presented in Thm 4.1 is not specific to concept models, but can be seen as a general statement on fidelity of performing dimensionality reduction with a linear random projection. It is not clear to me how NEC follows from here, as NEC is a measure of the sparsity of the final layer, which can be low even for large concept sets.

**Limitations:**

Yes, limitations have been addressed.

---

> ### Author Rebuttal · Authors · 2024-08-07
>
> Dear Reviewer MvHs,
>
> Thank you for the positive feedback. Please see below our responses to address your comments.
>
> **Q1:** It is not clear if NEC controls information leakage. As defined in [2], information leakage happens because of a `soft’ concept layer (that predicts probability of a concept instead of presence and absence). Though a small NEC reduces the possibility of information leakage and is a useful metric to check the goodness of the concept set.
>
> **A1:** Information leakage is a problem that haunts CBM models. As discussed in Sec 4 of [2], not only soft CBL has this problem, using hard concepts still shows information leakage, suggesting it’s hard to fully eliminate it in CBM. Though the reason for this has not been fully understood, we present Thm. 4.1 to provide a theoretical understanding of this problem. Additionally, in order to understand the real semantic information learned by CBL, we compare model performance with a random baseline as a “control group”, where no semantic information is learned. Our results in Figure 3 show that this random baseline drops significantly when NEC is low. This suggests that reducing NEC is a way to control information leakage.
>
> ---
>
> **Q2:** A study of time and compute complexity is needed, as every image has to be passed through a large object-detector. The effort to generate concepts is much higher (compared to [1], who generate concepts class wise).
>
> **A2:** Following your suggestion, we provide a comparative analysis of GPU-hours required to obtain annotations for three datasets: CUB, ImageNet, and Places365 in the table below. We would also like to clarify that we use concepts relevant to the class of an image when using Grounding-DINO. This helps reduce the false-positive rate of the open-vocabulary detection model and significantly decrease computation time. Please refer to Appendix D for more details.
>
> | Dataset           | GPU-hours |
> |-------------------|-----------|
> | CUB               | 0.5       |
> | ImageNet          | 110       |
> | Places365         | 138       |
>
> ---
>
> **Q3:** A study of the impact of proposed visually-grounded concept filtering would be interesting. For ex. What fraction of quantitative gains are because of the removed concept set? How much does augmentation contribute to overall performance gains?
>
> **A3:** We would like to clarify that concept filtering is a natural step in our training pipeline. An ablation here may not change the performance as it would be equivalent to training in a multi-label setting with certain labels never appearing in the training dataset.
>
> Further, following your request, we reported our provided an ablation study on data augmentation in the **Global Response #2.1.**
>
> ---
>
> **Q4:** The theoretical analysis, though appreciated, seems out of place. The result presented in Thm 4.1 is not specific to concept models, but can be seen as a general statement on fidelity of performing dimensionality reduction with a linear random projection. It is not clear to me how NEC follows from here, as NEC is a measure of the sparsity of the final layer, which can be low even for large concept sets.
>
> **A4:** Thank you for the comments, please allow us to clarify. Thm. 4.1 shows that the approximation error goes down linearly with the number of concepts, when CBL weights are randomly selected. This supports previous observations that a random CBL layer could also achieve good performance, suggesting the existence of information leakage. By applying NEC constraint, we actually constraint the number of concepts in Thm 4.1 (as other concepts do not contribute to the decision and could be ignored). Thus, the information leakage could be controlled as Thm. 4.1 suggested.
>
>  ---
>
> **Reference**
>
> [1] Oikarinen et al. “Label-Free Concept Bottleneck Models.” ICLR 2023
>
> [2] Mahinpei et al. “Promises and Pitfalls of Black-Box Concept Learning Models.” 2021
>
> ---
>
> **Summary**
>
> In summary,
>
> * In **Q1**, we discuss why controlling NEC could help control information leakage
>
> * In **Q2**, we provide a study on computational time and complexity.
>
> * In **Q3**, we provide an ablation study on the contribution of our data augmentation.
>
> * In **Q4**, we discuss how our theoretical result is connected to the NEC.
>
> In response to the weakness part, we have addressed the reviewer’s concern
>
> * in weakness #1 in **Q1**.
>
> * in weakness #2 by providing an analysis on computational time and complexity in **Q2**
>
> We believe that we have addressed all your concerns. Please let us know if you still have any reservations and we would be happy to address them!

---

### Official Review · Reviewer_QueL · 2024-07-16

**Soundness:** 2
**Presentation:** 3
**Contribution:** 2
**Rating:** 4
**Confidence:** 4

**Summary:**

This work proposed a new way to implement concept bottleneck models (CBMs), which use pretrained object detectors (Grounding-DINO) to filter and annotate the concepts. This step can make the concepts more visual and groundable, which improves the reliability of the concept predictor. This paper proposed a new metric (NEC) to control the number of concepts that the model uses when making the predictions. In the experiments of 5 datasets, the proposed model shows better performance using the NEC metric.

**Strengths:**

The idea of using the grounding model can help improve concepts' groundability. The paper is easy to follow in general.

**Weaknesses:**

1. **Limited contribution.** This paper is an incremental work to previous LLM-guided CBMs [1, 2] by adding a step to filter concepts with an object detection model. The findings that "CBM with random concepts can achieve good performance" have also been shown in previous works [3, 4]. From the technical part, the only difference between the proposed model and prior CBMs is the object detection model for concept filtering. However, the detected bounding boxes are not used in the model training and inference. Therefore, the technical contribution is also limited. Overall, this paper makes limited contributions to the community and does not provide enough insights for future work.

2. **Generalizability of the method.** The proposed method relies on an open-vocab detection model. However, open-vocab detection is still a very challenging task, and from my experience, the off-the-shelf models do not work well in many cases. For example, part-based detection is very difficult. In some datasets like DTD or action detection, the concepts are hard to capture in bounding boxes. Even in the evaluated datasets in this paper, like CIFAR, which is in low resolution, the performance of object detection is also quite skeptical. Although the authors have mentioned this limitation in the last section, I don't agree with their arguments that "prior work (e.g., LaBo, LM4CV, LF-CBM) also shared similar limitations on the reliance of CLIP". CLIP definitely has better generalization than the open-vocab detection model because the latter is a more challenging task. This concern is also reflected by the small set of datasets in their experiments, only 5 datasets, compared to what LaBo evaluated (11 datasets).

3. **Baseline Comparison.** The linear probing baseline is missing, which is necessary to justify the effectiveness of CBMs. For CUB, Place365, and ImageNet, the numbers of LM4CV and LaBo are missing. The reason shown in the caption that "they could not be applied on non-CLIP backbones" cannot convince me. I didn't see any difficulties in letting your model support CLIP. You can simply fine-tune CLIP using BCE loss in Eq (6) or just use CLIP as a feature extractor and learn a linear layer over it. In general, the numbers shown in Table 2 are quite low, far behind linear probing based on my experience. It seems the proposed method sacrifices a lot of performance for interpretability, which is not validated properly (see weakness 4). I feel that the performance gain of the proposed method comes from the priors of the object detection model.

4. **Human evaluation to justify faithfulness.** This paper claims that the proposed VLG-CBM is more interpretable and faithful. However, without concrete evidence like large-scale human evaluation, it is hard to justify VLG-CBM as more interpretable by just showing a few qualitative examples.

[1] Yang et al. Language in a Bottle: Language Model Guided Concept Bottlenecks for Interpretable Image Classification. CVPR 2023.
[2] Oikarinen et al. Label-Free Concept Bottleneck Models. ICLR 2023.
[3] Roth et al. Waffling around for Performance: Visual Classification with Random Words and Broad Concepts. ICCV 2023.
[4] Yan et al. Learning Concise and Descriptive Attributes for Visual Recognition. ICCV 2023.

**Questions:**

How is the threshold of detection confidence (T=0.15) chosen? Can this generalize to different datasets?

**Limitations:**

The authors discussed the limitations in Section 6.

---

> ### Author Rebuttal · Authors · 2024-08-07
>
> Dear Reviewer QueL,
>
> Thank you for the feedback, we believe there are some misunderstandings based on the comments. Please allow us to clarify below to address your comments.
>
> **Q1:** This paper is an incremental work to previous LLM-guided CBMs [1, 2] by adding a step to filter concepts with an object detection model.
>
> **A1:** We would like to clarify our contributions below and demonstrate that our contribution is much more than just filtering the concepts with object detection models. As described in the end of Sec 1, we have 3 main contributions spanning from a new methodology in sec 3, rigorous theoretical analysis in sec 4, to extensive evaluation in sec 5:
> 1. We are the first to propose an end-to-end pipeline for training CBM with annotations from grounding DINO and provide a method to utilize bounding boxes information obtained from the Grounding DINO, as shown in Sec 3.
> 2. As demonstrated and discussed in Sec 4, our work is the first to provide rigorous theoretical analysis for information leakage in CBMs and demonstrate that CBM can achieve optimal accuracy even on random concepts when the number of concepts is large enough
> 3. As discussed in Sec 4, we propose a new metric called the Number of Effective Concepts (NEC), which facilitates fair comparison between different CBMs by controling information leakage and helps improve interpretability of the CBMs.
> ---
> **Q2:** The findings that "CBM with random concepts can achieve good performance" have also been shown in previous works [3, 4]...
>
> **A2:** CBM with random concepts can achieve good performance as has been observed before, as you point out. However, we are the first to provide a theoretical analysis for it. As we prove in Thm. 4.1, The random concept CBL can approximate any linear function with approximation error that goes down linearly with the number of concepts, and reaches zero error when the number of concepts gets to the dimension of embedding.
>
> ---
> **Q3:** However, the detected bounding boxes are not used in the model training and inference. Therefore, the technical contribution is also limited…
>
> **A3:** We would like to clarify that the detected bounding boxes are indeed used during model training. As mentioned in Ln 134-136, we augment the training dataset by cropping images to a randomly selected bounding box and modifying the target one-hot vector to predict the concept corresponding to the bounding box.
>
> ---
>
> **Q4:** The proposed method relies on an open-vocab detection model. However, open-vocab detection is still a very challenging task, and from my experience, the off-the-shelf models do not work well in many cases. For example, …
>
> **A4:** We would like to highlight that the open-vocabulary object detectors primarily suffer from false-positive detections, and we address this issue in the following way:
>
> * Rather than using the entire concept set for detecting concepts in an image, we limit the concepts relevant to the ground truth class of the image. This ensures that the concepts have a high likelihood of being present in the image.
>
> * We used confidence threshold to filter bounding boxes with low confidence. In the Appendix D, we have evaluated concept annotations at different thresholds. We use CUB dataset for comparison which contains ground-truth for fine-grained concepts present in each image and report precision and recall metric to measure the quality of annotations from Grounding-DINO. As demonstrated in the appendix along with Table 2, Fig 4, Fig G.1 and G.2, the effect of false-positives is minimal and VLG-CBM is able to faithfully represent concepts in the CBL.
>
> ---
>
> **Q5:** This concern is also reflected by the small set of datasets in their experiments, only 5 datasets, compared to what LaBo evaluated (11 datasets).
>
> **A5:** Results on Flower102 and Food101: The Food-101 dataset achieved an ACC@NEC=5 of 81.68% and an Avg. Acc of 80.31%, while the Flower-101 dataset achieved higher scores with an ACC@NEC=5 of 90.58% and an Avg. Acc of 92.94% with CLIP-Resnet50 backbone.
>
> ---
>
> **Q6:**  The linear probing baseline is missing, which is necessary to justify the effectiveness of CBMs.
>
> **A6:** The linear probing results for Table 2 models are:
> 88.80% for CIFAR10, 70.10% for CIFAR100, 76.70% for CUB, 48.56% for Places and 76.13% for ImageNet. The gaps between our models (avg. acc) to the linear probing is 0.17% for CIFAR10, 3.62% for CIFAR100, 0.88% for CUB, 5.99% for Places and 2.15% for ImageNets.
>
> ---
>
>
> **Q7:** For CUB, Place365, and ImageNet, the numbers of LM4CV and LaBo are missing.
>
> **A7:** Following your request, we reported our results in the **Global Response #1 Additional models and datasets.** It can be seen that our VLG-CBM still outperforms the baselines under this setting.
>
> ---
>
> **Q8:** In general, the numbers shown in Table 2 are quite low, far behind linear probing based on my experience. It seems the proposed method sacrifices a lot of performance for interpretability, which is not validated properly (see weakness 4)...
>
> **A8:** First, we want to clarify that our accuracy is not far behind linear probing baseline, only 0-6% gap (refer to the numbers in A6) while other baselines have much larger gaps . In Table 2, we control NEC to be small (NEC=5) mainly for fairly compare different models: as shown in Figure 3, under dense setting (linear probing) even random CBL could achieve comparable performance to SOTA CBMs, which largely weaken its usefulness in comparing different CBMs.  In practice, users have the flexibility to choose the NEC to trade-off between performance and interpretability.
>
> ---
>
> **Q9:** …without concrete evidence like large-scale human evaluation, it is hard to justify VLG-CBM as more interpretable by just showing a few qualitative examples.
>
> **A9:** Following your suggestion, we have conducted additional large-scale human evaluations and demonstrated that our method outperformed the baselines. Please see the results in **Global Response #3 Human study.**

---

> > ### Comment · Reviewer_QueL · 2024-08-12
> >
> > Thanks for answering my questions. After reading your response and other reviewers' comments, I still believe this paper's contributions are limited. The main contritions of this paper are adding Grounding DINO in CBM and introducing a new metric. Adding object detection in CBM feels very trivial, and I don't think a new metric is a big contribution.

---

> > > ### Author Response · Authors · 2024-08-13
> > > **Clarification on misunderstanding of our contributions - Part 1**
> > >
> > > Dear Reviewer QueL,
> > >
> > > Thanks for the response! We appreciate the opportunity to further discuss our work and clarify a few more points that might have been misinterpreted or misunderstood. Based on the inaccurate descriptions of our contributions in the new comments, we are concerned that our work is undermined by the reviewer’s comments.
> > >
> > > Below we provide detailed explanations (#1-#4) of our main contributions to the field and discuss why they are both important and non-trivial. We believe these clarifications will effectively demonstrate the novelty and impact of our work.
> > >
> > > **#1.** First of all, there are **three** main contributions of our work instead of **two** that the reviewer mentioned, which we have already clarified in our rebuttal response **A1** that there are **three contributions** of our method:
> > > - **Contribution (1)**: The **first** end-to-end pipeline to build CBM with vision guidance from open-vocabulary object detectors in Sec 3;
> > > - **Contribution (2)**: The **first** rigorous theoretical analysis to prove that CBMs have serious issues on information leakage in Sec 4.1, whereas existing work [1, 2] only have empirical observations;
> > > - **Contribution (3)**: A **new** and **useful** metric to control information leakage problems that can facilitate fair comparison between different CBMs in Sec 4.2 and Sec 5.
> > >
> > > Notably, as we described in the introduction of the draft (see lines 34-42), existing CBMs still face two critical challenges that urgently needs to be fixed:
> > > - (a) inaccurate concept prediction problem, where the predicted concepts do not match the image and hurt the faithfulness of CBM;
> > > - (b) information leakage problem, where task-irrelevant concepts (or even random concepts) can produce a high accuracy CBMs, which also raise concerns on the faithfulness of the CBMs.
> > >
> > > Specifically, our **contribution (1)** is to address the problem of inaccurate concept prediction in (a) and our **contributions (2)&(3)** are to address the problem of information leakage problem in (b).
> > >
> > > Together with our 3 contributions, we show in extensive evaluations in sec 5 (see Table 2, 3, Figure 4), and global response in the rebuttal (additional experiments, ablation study and human study) that our VLG-CBM framework is a promising first step towards building a faithful, reliable, and high performance CBM, which is currently lacking in the field and literature. More specifically, our method provides more favorable explanations than the baselines by around 18% in human study, our method also achieves up to 29.78% improvement on average accuracy and gets up to 51.09% improvement on accuracy at NEC=5.
> > >
> > > ---
> > >
> > > **#2**. Why is our **Contribution (1)** non-trivial and important to the field?
> > >
> > > As we clarified in the above point **#1**, our **Contribution (1)** is the **first** end-to-end pipeline to build CBM with vision guidance from open-vocabulary object detectors described in Sec 3.
> > >
> > > We respectfully disagree with the reviewer’s comment that “*adding object detection in CBM feels very trivial*”. To our best knowledge, there is no other work in the CBM literature that has the idea of leveraging object detection to address the concept faithfulness problem in CBMs. We are the first to provide this elegant and effective approach to address the inaccurate concept prediction problems.
> > >
> > > Besides, the reviewer mentioned that using the open vocabulary detection model is very challenging in **Q4**, which exactly indicates that it is non-trivial to apply open-vocabulary object detectors directly to the pipeline. As we responded in **A4** of our rebuttal response, we pointed out that open-vocabulary object detectors could indeed suffer from false-positive detections and we described how we have addressed this challenge in our pipeline.

---

> > > > ### Author Response · Authors · 2024-08-13
> > > > **Clarification on misunderstanding of our contributions - Part 2**
> > > >
> > > > To give more details, our work encompasses the following additional crucial steps in the pipeline and have not been studied previously in the context of CBMs:
> > > > 1. **Handling False-positives from foundational models**: False-positive outputs from foundational vision models currently limit their use in training CBMs. Our work provides a simple, yet clever, way to solve the issues with false positive detections: i) We use concepts corresponding to a class for obtaining annotation for an image. ii) We use confidence threshold to remove bounding boxes with low confidence. As demonstrated in the appendix D along with Table 2, Fig 4, Fig G.1 and G.2, and the ablation study on threshold in Table C.1 in appendix C, this design significantly reduces the effect of false-positives, making foundational vision models viable for training CBMs
> > > > 2. **Using obtained bounding boxes for training**: We use the generated bounding box for training the CBL layer. As mentioned in Ln 134-136 in our draft, we augment the training dataset by cropping images to a randomly selected bounding box and modifying the target one-hot vector to predict the concept corresponding to the bounding box. We have provided an ablation study highlighting the effectiveness of this approach in global response under #2 ablation studies.
> > > >
> > > > ---
> > > >
> > > > **#3**. Why is our **Contribution (2)** novel and important to the field?
> > > >
> > > > As we clarified in the above point **#1**, our **Contribution (2)** is the **first** rigorous theoretical analysis to prove that CBMs have serious issues on information leakage in Sec 4.1, whereas existing work [1] only has empirical observations.
> > > >
> > > > As we pointed out in the original manuscript (Line 40-42, Line 152-162), information leakage is an important issue that was observed by prior work [1, 2]. However, there is no rigorous theoretical understanding of this phenomenon in [1, 2]. As we responded in our **A2** in the rebuttal response, we are the first work to provide the first theoretical analysis to explain this phenomenon and therefore identify the number of concepts as an important factor of information leakage in Thm. 4.1, which inspired us to control information leakage via the novel NEC metric that we propose in Sec 4.2 (which is our **Contribution (3)**).
> > > >
> > > > ---
> > > >
> > > > **#4.** Why is our **Contribution (3)** novel and important to the field?
> > > >
> > > > As we clarified in the above point **#1**, our **Contribution (3)** is a **new** and **useful** metric to control information leakage problems that can facilitate fair comparison between different CBMs in Sec 4.2 and Sec 5.
> > > >
> > > > We respectfully disagree with the reviewer's comments that *"I don't think a new metric is a big contribution"*. As we described in the introduction of the draft (see lines 34-42), existing CBMs have serious problems of faithfulness due to information leakage problems. Without this metric, the performance of CBMs do not correctly reflect their "true" performance, which causes a false sense of confidence. Indeed, as we show in Table 2 in the draft and Table R1 in the Global response in the rebuttal, under the fair comparison setting, many existing CBMs' true performance are much worse than what they were reported. For example, using Acc@NEC=5, LF-CBM has around 2%-21% drop, Labo has around 20%-58% accuracy drop, LM4CV has around 34%-63% accuracy drop than reported, providing a false sense of performance.
> > > >
> > > > Additionally, we would like to highlight that our proposed new metric NEC is a theoretically grounded solution to control information leakage problems and has much more flexibility in concept choices and better interpretability compared with the previous approach [1]. In [1], the authors proposed to control the total number of concepts, which however limits the choice of concepts and can harm the interpretability due to dense connection, as supported by the human study results in **General Response #3** and the qualitative example in **Fig. 1** and **Fig. H.2 - H.4** in the appendix of our draft.
> > > >
> > > > ---
> > > >
> > > > ## Summary
> > > > To sum up, we believe that there are misunderstandings from the reviewer on our contributions and novelty due to inaccurate description. To clarify the misunderstanding, in this response, we address the reviewer's additional comments by
> > > > * Clarifying our 3 contributions in **#1**
> > > > * Clarifying the importance and novelty of our **Contribution (1)** in **#2**
> > > > * Clarifying the importance and novelty of our **Contribution (2)** in **#3**
> > > > * Clarifying the importance and novelty of our **Contribution (3)** in **#4**
> > > >
> > > > Please let us know if you still have any concerns, we would be happy to discuss further.
> > > >
> > > >
> > > > ## Reference
> > > > [1] An et al. "Learning concise and descriptive attributes for visual recognition." ICCV 23
> > > >
> > > > [2] Mahinpei et al. "Promises and pitfalls of black-box concept learning models." ICML 21

---

> > > > > ### Author Response · Authors · 2024-08-14
> > > > > **Follow-up and Additional results to show the generalizability of our methods**
> > > > >
> > > > > Dear Reviewer QueL,
> > > > >
> > > > > As the rebuttal period is close to an end, please let us know if you still have any remaining concerns. We have tried to address all your concerns in our rebuttal response and we also give additional clarification on our contributions to address your new comments yesterday.
> > > > >
> > > > > While we are waiting for your follow-up response, we would like to share additional results on four new datasets to further address your concerns in Weakness #2 and #3. In the rebuttal, we have provided some of the additional experiments results in **A5** as requested and now we would like to provide more results to further support the generalizability of our methods.
> > > > >
> > > > > **1. Additional results on 4 more datasets**
> > > > >
> > > > > Specifically, we provide additional results on four new datasets that Labo [1] used: Aircraft, Flower102, Food101, and DTD. For a fair comparison of our method with  LaBo[1], we have followed the same dataset splits as used by LaBo and reported accuracy for both LaBo and VLG-CBM on the Resnet50-CLIP backbone. The results of our experiments are reported in the table below. We observe that VLG-CBM consistently outperforms LaBo on Aircraft, Flower102, Food101, and DTD dataset by as much as 8.16% points on average accuracy, and 12.72% on Acc@5.
> > > > >
> > > > > This demonstrates that our method can successfully generalize to fine-grained and challenging classification datasets (e.g. DTD as mentioned by the reviewer) and outperform existing state-of-the-art CBMs like Labo. Additionally, our original result on Cifar 10 and Cifar 100 also outperforms the baselines by a large margin (e.g. on Acc accuracy, our method is 3-19% better on Cifar 10, and 4-29% better on Cifar 100) as shown in the Table 2 in our manuscript. Furthermore, on larger dataset like Imagenet, it’s 0.5-36% better on the Avg accuracy, as shown in the Table R1 in the global response of the rebuttal.
> > > > >
> > > > > Table R4: Comparison on additional 4 datasets
> > > > >
> > > > > | Dataset   |     Acc@5    |        |   Avg. Acc   |        |
> > > > > |-----------|:------------:|:------:|:------------:|:------:|
> > > > > |           | Ours(VLGCBM) | LaBo   | Ours(VLGCBM) | LaBo   |
> > > > > | Aircraft  | 36.54%       | 23.82% | 39.26%       | 31.10% |
> > > > > | Flower102 | 94.19%       | 81.89% | 95.36%       | 93.06% |
> > > > > | Food101   | 78.02%       | 66.46% | 79.58%       | 75.05% |
> > > > > | DTD       | 67.67%       | 60.40% | 69.82%       | 66.37% |
> > > > >
> > > > > Note: The results in **A5** were generated for Food and Flower dataset following splits from the pytorch library. To have a fair comparison with LaBo, we follow LaBo’s split as described below, which is different from the standard split in the pytorch library used in **A5**, resulting in a slight difference in the result number.
> > > > >
> > > > > | Dataset | Pytorch-test-size | Labo-test-size |
> > > > > |---------|-------------------|----------------|
> > > > > | Food    | 25,250| 30,300 |
> > > > > | Flower  | 1,020 | 2,463|
> > > > >
> > > > > The above results will be included in the revised manuscript.
> > > > >
> > > > > ---
> > > > >
> > > > > **2. To sum up, we believe that we have addressed all your concerns in the 4 Weaknesses points in the review comments with the extensive evaluations, additional requested experiments, and clarifications.**
> > > > >
> > > > > Specifically,
> > > > >
> > > > > * The **weakness point #1** on **Limited contribution** is addressed by our clarification response yesterday (Aug 12) with title “Clarification on misunderstanding of our contributions - Part 1” and “Clarification on misunderstanding of our contributions - Part 2”. We clarified that there are 3 main contributions (the first end-to-end pipeline, the first theoretical analysis on CBM leakage with random concepts, and a new and useful metric for fair comparison between CBM) and explain why each of them are non-trivial and novel to the field.
> > > > > * The **weakness point #2** on **Generalizability of the method** is addressed by our rebuttal and additional result, which shows that our methods generalize to the DTD and CIFAR dataset that the reviewer mentioned to be challenging and outperform existing methods.
> > > > > * The **weakness point #3** on **Baseline Comparisons** is addressed by our rebuttal and additional results, because we have shown that our methods outperform existing methods on numerous datasets, including Cifar 10, Cifar 100, ImageNet, CUB, Place365, Flower 102, Food 101, DTD, Aircraft, as demonstrated in Table 2 in the manuscript, Table R2 in the global response, and Table R4 in this response.
> > > > > * The **weakness point #4** on **Human evaluation to justify faithfulness** is addressed by our rebuttal with human study in the global response of rebuttal (Table R3) where we show that our methods outperforms the baselines in the human study.
> > > > >
> > > > > We value the reviewer’s input and welcome any additional comments. If our clarifications and additional results address your major concerns, we would appreciate your consideration in raising the score. Thanks for your time and feedback.
> > > > >
> > > > > ---
> > > > >
> > > > > ## Reference
> > > > >
> > > > > [1] Yang etal. Language in a bottle: Language model guided concept bottlenecks for interpretable image classification. CVPR 23.

---

### Official Review · Reviewer_UnrN · 2024-07-22

**Soundness:** 3
**Presentation:** 3
**Contribution:** 2
**Rating:** 5
**Confidence:** 3

**Summary:**

The paper introduces Vision-Language-Guided Concept Bottleneck Model (VLG-CBM), an innovative approach to training Concept Bottleneck Models (CBMs) using vision-language models (VLMs). This method aims to improve the faithfulness and performance of CBMs by addressing the limitations of existing models, specifically inaccurate concept predictions and information leakage. VLG-CBM leverages grounded object detectors for visually recognizable concept annotations and introduces a new metric, the Number of Effective Concepts (NEC), to control information leakage and enhance interpretability. The method demonstrates significant performance improvements across multiple benchmarks.

**Strengths:**

1) The paper is well written with a clear logical flow.

2) The approach automates the creation of concept datasets using grounded object detectors, which means there's no need for manual annotations. This saves time and effort, making the method easier to scale up for larger datasets. Plus, it reduces the chances of human error in the data, leading to more reliable training results.

3) The paper provides theoretical explanations for information leakage in Concept Bottleneck Models (CBMs). By introducing the Number of Effective Concepts (NEC) metric, the authors offer a way to control and measure information leakage, enhancing the overall integrity and interpretability of the model.

3) The VLG-CBM model shows significant improvements in accuracy across five standard benchmarks, with gains ranging from 2.4% to 7.6% over current methods.

**Weaknesses:**

1）The proposed pipeline integrates several independent models, ie, LLM for concept candidate generation, and GroundingDINO for bounding box detection. I'm wondering how well the grounding detector can do in discovering the concept candidate.  And if the detector fails to detect a key concept, will the final recognition result be affected?

2) Dependence on Pre-trained Models: The reliance on pre-trained VLMs and object detectors may limit the approach's applicability to domains where such models are not readily available or effective.

3) My major concern is in generalization. While the method shows impressive results on standard image recognition benchmarks, its effectiveness in more diverse or less-structured real-world scenarios needs further validation. For example, how would the VLG-CBM perform for OOD image recognition like NICO or Water-Bird, or how could it handle more complex tasks like VQA?

**Questions:**

see weaknesses

**Limitations:**

see weaknesses

---

> ### Author Rebuttal · Authors · 2024-08-07
>
> Dear Reviewer UnrN,
>
> Thank you for the positive feedback! Please see our responses below to address your comments.
>
> ---
> **Q1:** How well the grounding detector can do in discovering the concept candidate. And if the detector fails to detect a key concept, will the final recognition result be affected?
>
> **A1:** In Appendix D, we provided a study evaluating concept annotations obtained from Grounding-DINO. We use CUB dataset for comparison which contains ground-truth for fine-grained concepts present in each image and report precision and recall metric to measure the quality of annotations from Grounding-DINO.
>
> The confidence threshold controls when a concept will be used in training the model. It is possible that a concept is present with a confidence score below threshold in an image and will  not be used during training. Please see the ablation study for this quantity in Appendix C.1.
>
> ---
>
> **Q2:** The reliance on pre-trained VLMs and object detectors may limit the approach's applicability to domains where such models are not readily available or effective.
>
>
> **A2:** We would like to clarify that this is a general drawback with the use of foundational models and most existing CBMs [1-3] use foundation models to remove the need for human annotation in various phases of the CBM pipeline. For example, LF-CBM[1] uses CLIP model for assigning score to concepts, and existing methods like LaBO[2], LM4CV[3], and LF-CBM use LLMs for generating concept sets.
>
> ---
>
> **Q3:** How would the VLG-CBM perform for OOD image recognition like NICO or Water-Bird, or how could it handle more complex tasks like VQA?
>
> **A3:** Following your suggestion, we have conducted additional experiments to evaluate how VLG-CBM performs in OOD datasets. We trained our VLG-CBM on the CUB dataset and tested it on the Water-Bird dataset to evaluate the OOD performance. The Water-bird dataset is constructed by cropping out birds from the CUB dataset and transferring them onto backgrounds from the Places dataset. We compare the performance of VLG-CBM with the standard black-box model trained on the CUB dataset. It can be seen that our VLG-CBM generalizes well as the standard model does, which shows that our VLG-CBM is competitive and has very small accuracy trade-off with the interpretability compared with the standard black-box model.
>
> | Method                     | CUB-Acc | waterbird-Acc |
> |----------------------------|---------|---------------|
> | Standard model (black-box) |  76.70% |        69.83% |
> | VLG-CBM                    |  75.79% |        69.83% |
>
> For the question on more complex tasks like VQA, it is a very different task than the image classification task that current CBMs [1-3] are designed for, as a typical model for VQA task would involve the image embedding and question embeddings. Nevertheless, we think that our VLG-CBM can be potentially useful to convert the non-interpretable image embedding to an interpretable embeddings through a concept bottleneck layer. We will include this interesting question to future work and add a discussion in the revised draft.
>
> ---
>
> **Reference**
>
> [1] Oikarinen etal. Label-free concept bottleneck models. ICLR 23
>
> [2] Yang etal. Language in a bottle: Language model guided concept bottlenecks for interpretable image classification. CVPR 23.
>
> [3] Yan etal. Learning concise and descriptive attributes for visual recognition. ICCV 23.
>
> ---
>
> **Summary**
>
> In summary,
>
> * In **Q1**, we added discussion on the performance of the grounding detector.
>
> * In **Q2**, we respond to the concern on reliance on pretrained VLMs.
>
> * In **Q3**, we conducted an extensive study on the waterbird dataset to show generalizability of our model to OOD datasets.
>
> In response to the weakness part, we have addressed the reviewer’s concern
>
> * in weakness #1 in **Q1**.
>
> * in weakness #2 by comparing with other baselines in **Q2**.
>
> * in weakness #3 by adding an extensive study on the waterbird dataset in **Q3**.
>
> We believe that we have addressed all your concerns. Please let us know if you still have any reservations and we would be happy to address them!

---

### Official Review · Reviewer_hsJW · 2024-07-23

**Soundness:** 3
**Presentation:** 3
**Contribution:** 3
**Rating:** 5
**Confidence:** 3

**Summary:**

This paper proposed a framework that leverages existing grounding models to refine the accurate concept annotations, and therefore enables better CBM training. Also, a new metric is proposed to avoid leakage and improve interoperability. Experiments and visualizations are conducted to demonstrate the superior performance of the proposed method.

**Strengths:**

1. For methodology, incorporating a grounding model to get faithful concepts, and proposing a metric - Number of Effective Concepts to control the information leak and supervise sparse layer optimization are interesting and novel to a certain degree in this field.

2. The proposed method outperforms previous approaches in multiple widely used benchmarks.

**Weaknesses:**

1. Missing comparison with recent works in CMB, for example,  [1] and [2].

2. How good is LLM at generating candidate concepts, and how the different LLMs' quality would impact the generated candidate quality as well as final performance? An ablation/analysis is needed.

3. How good is the grounding model in terms of object class matching and box location prediction? The ablation of different grounding models and even a random baseline should be added. I am also curious about the training augmentation based on bounding box, how much gain it can bring?

4. It's natural that different classes have different numbers of concepts to distinguish the class because some are complex and some are simple. If forcing the NEC to certain number, would the model automatically learn to assign different numbers of concepts to different classes? An analysis is needed.

Refs:\
[1]. Pham, Thang M., et al. "PEEB: Part-based Image Classifiers with an Explainable and Editable Language Bottleneck." arXiv preprint arXiv:2403.05297 (2024). \
[2]. Sun, Ao, et al. "Eliminating information leakage in hard concept bottleneck models with supervised, hierarchical concept learning." arXiv preprint arXiv:2402.05945 (2024).

**Questions:**

In summary, my concerns mainly focus on the lack of comprehensive comparison, ablation, and analysis.

**Limitations:**

Yes, it is discussed in Section 6.

---

> ### Author Rebuttal · Authors · 2024-08-07
>
> Dear Reviewer hsJW,
>
> Thank you for the positive feedback! Please see our responses below to address your comments.
>
> **Q1**. Missing comparison with recent works in CMB, for example, [1] and [2]
>
> **A1**: Thank you for providing the reference [1, 2]. Below we discuss the differences between our method and [1] and [2]
> - [1]: Similar to us, [1] also uses an open-vocabulary object detection model to provide an explainable decision. However, their model is directly adapted from an OWL-ViT model, while our VLG-CBM uses an open-vocabulary object detection model to train a CBL over any base model, providing more flexibility. Additionally, their model requires pretraining to get best performance, while our VLG-CBM could be applied post-hoc to any pretrained model.
> - [2]: This paper also aims at eliminating the information leakage. They evaluate it by measuring the performance drop speed after removing top-contributing concepts, which can be controlled by our proposed NEC metric. This is because the performance should reach minimum after removing all contributing concepts.
>
> We will cite the papers [1,2] and discuss the differences in the revised manuscript.
>
> [1]. Pham, Thang M., et al. "PEEB: Part-based Image Classifiers with an Explainable and Editable Language Bottleneck." arXiv preprint arXiv:2403.05297 (2024).
>
> [2]. Sun, Ao, et al. "Eliminating information leakage in hard concept bottleneck models with supervised, hierarchical concept learning." arXiv preprint arXiv:2402.05945 (2024)
>
> ---
>
> **Q2**: How good is LLM at generating candidate concepts, and how the different LLMs' quality would impact the generated candidate quality as well as final performance?
>
> **A2**: We would like to clarify that the focus of this work is on proposing a novel CBM training pipeline by grounding concepts with open-domain object detectors and not on obtaining the concept set. As mentioned in Ln 110-112, we use the concept set from LF-CBM for training VLG-CBM and this could be replaced with concept sets obtained from other methods.
>
> ----
>
> **Q3**: How good is the grounding model in terms of object class matching and box location prediction?
>
> **A3**: In the Appendix D, we have provided a study evaluating concept annotations obtained from Grounding-DINO. We use CUB dataset for comparison which contains ground-truth for fine-grained concepts present in each image and report precision and recall metric to measure the quality of annotations from Grounding-DINO. As demonstrated in the appendix along with Table 2, Fig 4, Fig G.1 and G.2, the effect of false-positives is minimal and VLG-CBM is able to faithfully represent concepts in the CBL.
>
> ---
>
> **Q4**: The ablation of different grounding models and even a random baseline should be added.
>
> **A4**: Following your suggestion, we have conducted additional experiments and provided an ablation study with two different versions of GroundingDINO: Swin-T and Swin-B, where Swin-B is a larger model and has better open-vocabulary detection performance. We observe an increase in accuracy when using Swin-B for both NEC=5 and avg accuracy metric. This shows the potential of our method to scale as the performance of open-domain object detectors continues to improve.
>
> For the random baseline, we have provided results in Table 2 in the original draft by starting with a randomly initialized CBL layer and training a linear layer with our sparsity constraints. This would be equivalent to randomly assigning concepts from the concept set to each image and training the final layer.
>
> | DINO-Model | Acc@NEC=5 | Avg. Acc |
> |------------|-----------|----------|
> | Swin-T     | 75.30     | 75.54    |
> | Swin-B     | 75.79     | 75.82    |
>
>
> ---
>
> **Q5**: Gains with training augmentation based on bounding box
>
> **A5**: Following your request, we reported our results in the Global Response #2.1.
>
> ---
>
> **Q6**: If forcing the NEC to certain number, would the model automatically learn to assign different numbers of concepts to different classes? An analysis is needed.
>
> **A6**: In Appendix E, we have shown the distribution of non-zero weight on CUB and Places365 models. As the Fig E.1 shows, our method automatically learns to use a different number of concepts for classifying different classes.
>
> ---
>
> **Summary**
>
> In summary,
> - In **Q1**, we discussed and compared with recent works [1] and [2]
> - In **Q2**, we answered the question on concept set generation.
> - In **Q3**, we discussed the effect of grounding models on concept annotations
> - In **Q4**, we provided an ablation study on the grounding model used in VLG-CBM.
> - In **Q5**, we added an ablation study to study the impact of our augmentation probability.
> - In **Q6**, we answered the question on the distribution of non-zero weights between different classes.
>
> In response to the weakness part, we have addressed the reviewer’s concerns
> - in weakness #1 by providing a comparison in **Q1**.
> - in weakness #2 in **Q2**
> - in weakness #3 by adding ablation study on object detection models and augmentation based on bounding box in **Q3** and **Q4**
> - in weakness #4 in **Q6**
>
> We believe that we have addressed all your concerns. Please let us know if you still have any reservations and we would be happy to address them!

---

> > ### Author Response · Authors · 2024-08-13
> > **Additional results for the quality of concept sets in Q2**
> >
> > Dear Reviewer hsJW,
> >
> > We would like to provide additional results regarding your concerns in **Q2** on the quality of concept sets obtained from different LLMs. We have conducted an ablation study to obtain concept sets from two additional LLMs (LLama3-8B and GPT4o-Mini) following methodology in [1].
> >
> > The results of our experiments are reported in the table below. We observe that the results are similar across three different LLMs, with a slight deviation within 0.5%. Specifically, we find that GPT4oMini performs the best, achieving a maximum average accuracy of 76.11% on the CUB dataset. We will include this ablation study to the revised draft.
> >
> > Please let us know if you have any additional questions or concerns, and we would be happy to resolve them!
> >
> > |               | Acc@NEC=5 | Avg. Acc |
> > |---------------|-----------|----------|
> > | CUB_GPT3      | 75.79     | 75.82    |
> > | CUB_LLama3    | 75.57     | 75.73    |
> > | CUB_GPT4oMini | 76.07     | 76.11    |
> >
> > ---
> >
> > ## Reference
> > [1] Oikarinen et al. Label-free concept bottleneck models. ICLR 23

---

### Author Rebuttal · Authors · 2024-08-07

**General response: New Experiments**

Thank you for all the thoughtful reviews. In response, we have performed many experiments during the rebuttal period as requested by the reviewers, mainly to evaluate our method on more model architectures and datasets, as well as several ablation studies to understand the importance of each part in our pipeline.

1. **Additional models and datasets**:

- **We add CLIP-RN50 backbone results to compare with LaBo and LM4CV in CUB and ImageNet datasets**.
The results show that our method consistently outperformed both LaBo and LM4CV.
Table R1: Performance comparison of CLIP-RN50 backbone on all baselines
  | Method       | ImageNet  | ImageNet  | CUB    | CUB        |
  |--------------|----------------|-------------------|-----------|--------------|
  |                     | Acc@5      | Avg. Acc   | Acc@5 | Avg. Acc |
  | VLG-CBM (ours) | **59.74%**         | **62.70%**            | **60.38%**    | **66.03%**       |
  | LF-CBM       | 52.88%         | 62.24%            | 31.35%    | 52.70%       |
  | LaBo         | 24.27%         | 45.53%            | 41.97%    | 59.27%       |
  | LM4CV        | 3.77%          | 26.65%            | 3.63%     | 15.25%       |

- **We add experiments on Waterbird dataset to show generalizability to OOD datasets**:  In order to see how VLG-CBM performs in OOD datasets, we train our VLG-CBM on the CUB dataset and test our VLG-CBM on Water-Bird dataset to evaluate the OOD performance. The Water-bird dataset is constructed by cropping out birds from the CUB dataset and transferring them onto backgrounds from the Places dataset. We compare the performance of VLG-CBM with the standard black-box model trained on the CUB dataset. It can be seen that our VLG-CBM generalizes well as the standard model does, which shows that our VLG-CBM is competitive and has very small accuracy trade-off with the interpretability compared with the standard black-box model.
Table R2: OOD Performance of VLG-CBM
  | Method                        | CUB-Acc | Waterbird-Acc |
  |-------------------------------|---------|---------------|
  | Standard model (black-box)    | 76.70%  | 69.83%        |
  | VLG-CBM                       | 75.79%  | 69.83%        |

2. **Ablation Studies**

- **Data Augmentation**: To understand the effect of augmentation, we provide an ablation study on the CUB dataset in the table below. We observe that a crop-to-concept-prob of 0.2 works best for the CUB dataset, and augmentation improve the accuracy. We will provide an ablation study on remaining datasets in the revised manuscript.
  | Crop-to-Concept-Prob | Acc@NEC=5 | Avg. Acc |
  |----------------------|-----------|----------|
  | 0.0                  | 75.73     | 75.76    |
  | 0.2                  | 75.83     | 75.88    |
  | 0.4                  | 75.71     | 75.72    |
  | 0.6                  | 75.57     | 75.62    |
  | 0.8                  | 75.52     | 75.57    |
  | 1.0                  | 72.29     | 73.15    |

3. **Human study**
  We conduct a human study following the approach of LF-CBM on Amazon MTurk, showing the annotator top-5 contributing concepts of our method (VLG-CBM) and baseline (LF-CBM or LM4CV) and asking them which one is better. The scores for each method are assigned as 1-5 according to the response of annotators: 5 for the explanations from VLG-CBM is strongly more reasonable, 4 for VLG-CBM is slightly more reasonable, 3 for both models are equally reasonable, 2 for the baseline is slightly more reasonable, and 1 for the baseline is strongly more reasonable. Thus, if our model provides better explanations than the baselines, then we should see a score higher than 3.
We report the average score in below table for two baselines: LF-CBM and LM4CV. It can be seen that VLG-CBM has scores higher than 3 for both baselines, indicating our VLG-CBM provides better explanations than both baselines.
  LaBo is excluded in our experiment due to its dense layer and large number of concepts: the top-5 concepts usually account for less than 0.01% of final prediction. We provide an example in the supplementary pdf to show this.
  Table R3: Human study result (standard deviation in parentheses)
  | Experiment                | Score (VLG-CBM) | Score (Baseline) |
  |---------------------------|-----------------|------------------|
  | VLG-CBM vs. LF-CBM        | 3.33 (1.54)     | 2.67 (1.54)      |
  | VLG-CBM vs. LM4CV         | 3.38 (1.54)     | 2.62 (1.54)      |

---

### Comment · Area_Chair_bET4 · 2024-08-12

Dear Reviewers,

Thanks for your efforts in reviewing this paper again! The authors have provided a detailed rebuttal to answer your concerns and questions. Please take a look, and let us (PCs, SACs, ACs, peer reviewers, and authors) know whether your concerns have been addressed. If not, please highlight your remaining concerns. And the authors still have opportunities to answer your questions.

Best,

AC

---

### Decision · Program_Chairs · 2024-09-25

**Decision:**

Accept (poster)

**Comment:**

After the rebuttal, the submission received relatively positive reviews (1 Accept, 3 Borderline Accept, and 1 Borderline Reject). The main concerns about the papers are: 1) The novelty of the proposed idea is limited. 2) More comprehensive studies are expected. During the rebuttal, the authors provided extensive new results to show the generalization ability of the proposed method. Overall, I think the submission is worth to be published and recommend **Accept** for this submission. As suggested by reviewers, in the camera-ready version, a more detailed comparison with previous works should be added.